# Atomic models of the *Toxoplasma* cell invasion machinery

Jianwei Zeng [1,7], Yong Fu [2,6,7], Pengge Qian [2,7], Wei Huang [3], Qingwei Niu[1,4], Wandy L. Beatty[2], Alan Brown [5], L. David Sibley [2] ✉ & Rui Zhang [1] ✉

Apicomplexan parasites, responsible for toxoplasmosis, cryptosporidiosis and malaria, invade host cells through a unique gliding motility mechanism powered by actomyosin motors and a dynamic organelle called the conoid. Here, using cryo-electron microscopy, we determined structures of four essential complexes of the *Toxoplasma gondii* conoid: the preconoidal P2 ring, tubulin-based conoid fibers, and the subpellicular and intraconoidal microtubules. Our analysis identified 40 distinct conoid proteins, several of which are essential for parasite lytic growth, as revealed through genetic disruption studies. Comparative analysis of the tubulin-containing complexes sheds light on their functional specialization by microtubule-associated proteins, while the structure of the preconoidal ring pinpoints the site of actin polymerization and initial translocation, enhancing our mechanistic understanding of gliding motility and, therefore, parasite invasion.

The eukaryotic phylum of Apicomplexa includes several important human parasites, such as *Toxoplasma gondii*, *Cryptosporidium* spp. and *Plasmodium* spp., which are responsible for toxoplasmosis, cryptosporidiosis and malaria, respectively. These intracellular parasites invade host cells and tissues using a unique mechanism of gliding motility[1,2], which is powered by actomyosin motors[3–5] and relies on a dynamic, cone-shaped organelle known as the conoid[6].

The *T. gondii* conoid (Fig. 1a) is a cage made up of ~14 spiral conoid fibers (CFs), each formed by nine tubulin protofilaments arranged in a bent C-shaped open tubule[7]. These CFs are packed against each other and are capped at the apical end by three preconoidal rings (PCRs) and at their base by an apical polar ring (APR), through which the conoid extrudes and retracts[8,9]. Within the conoid cage are a pair of conventional microtubules made of 13 protofilaments[7]. These intraconoidal microtubules (ICMTs) have an essential role in the docking and discharge of secretory organelles termed rhoptries[10], the contents of which are essential for host cell invasion[11,12]. In *Cryptosporidium parvum*

sporozoites, the conoid is shorter but key features such as the PCRs and APR are conserved[13]. The conoid structure is even more reduced in *Plasmodium* spp., although the PCRs and APR remain conserved and a short, flattened conoid structure is observed in the motile ookinete stage within the mosquito midgut[14–17].

The APR in *T. gondii* anchors the minus ends of 22 subpellicular microtubules (SPMTs) that extend posteriorly to subtend the membrane along two thirds of the cell body (Fig. 1a). These SPMTs are associated with the inner membrane complex (IMC), a set of flattened membrane vesicles (alveoli) that lie beneath the parasite's plasma membrane[18]. Unlike mammalian microtubules, SPMTs in *T. gondii* are remarkably stable, retaining their structure under detergent treatment and cold exposure[19,20]. This stability is mainly attributed to the binding of microtubule inner proteins (MIPs)[21,22] and the integral membrane proteins of the IMC[23].

Our current knowledge about the protein composition of the conoid has been obtained through microscopy of epitope-tagged proteins[6,15,24–26]

[1]Department of Biochemistry, Washington University in St. Louis, School of Medicine, St. Louis, MO, USA. [2]Department of Molecular Microbiology, Washington University in St. Louis, School of Medicine, St. Louis, MO, USA. [3]Department of Pharmacology, Case Western Reserve University, Cleveland, OH, USA. [4]Molecular Cell Biology (MCB) Graduate Program, Division of Biology & Biomedical Sciences, Washington University in St. Louis, School of Medicine, St. Louis, MO, USA. [5]Department of Biological Chemistry and Molecular Pharmacology, Blavatnik Institute, Harvard Medical School, Boston, MA, USA. [6]Present address: State Key Laboratory of Veterinary Public Health and Safety, College of Veterinary Medicine, China Agricultural University, Beijing, China. [7]These authors contributed equally: Jianwei Zeng, Yong Fu, Pengge Qian. ✉e-mail: sibley@wustl.edu; zhangrui@wustl.edu

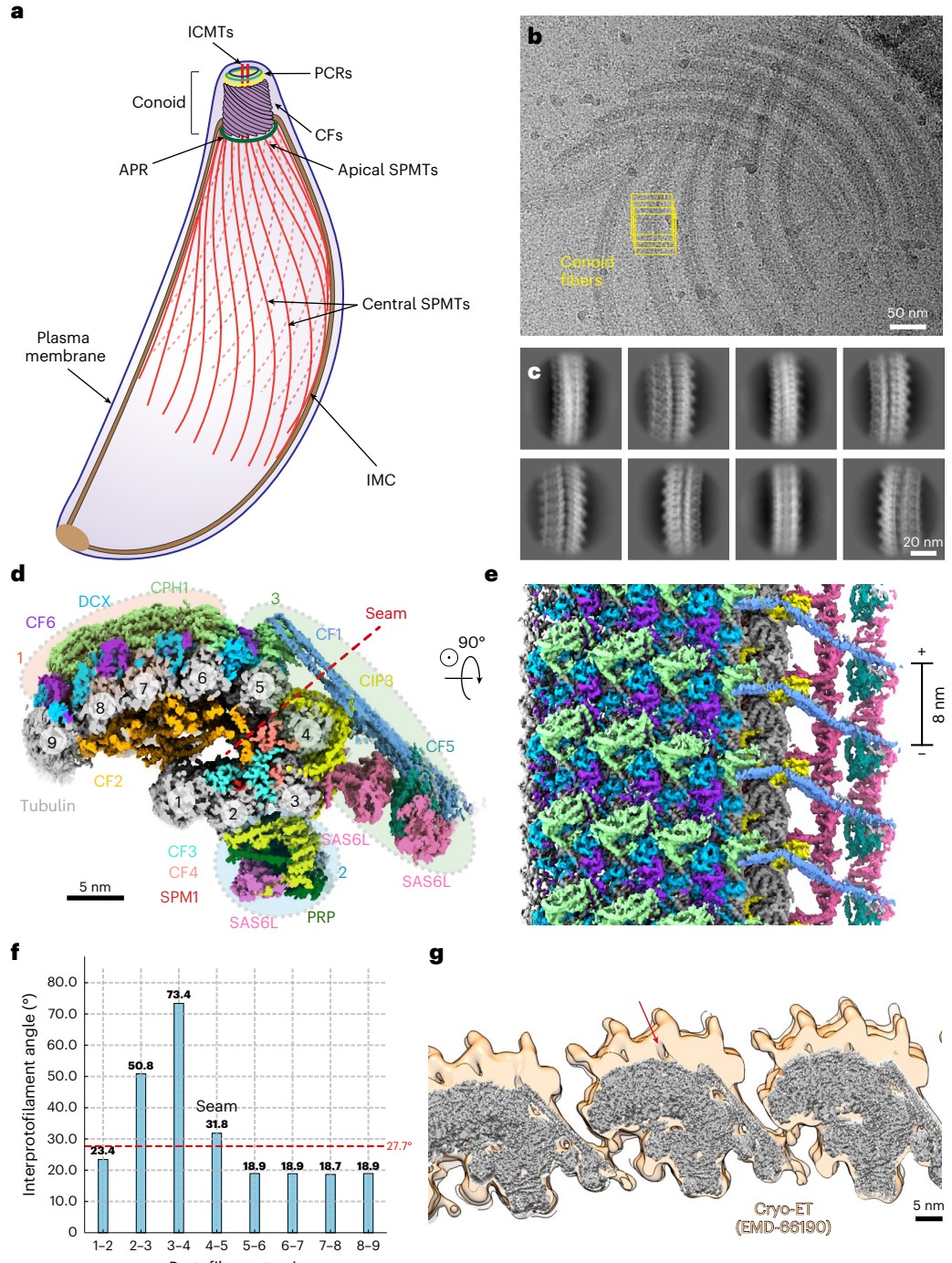

**Fig. 1 | Cryo-EM structure of the CF. a**, Schematic diagram of *T. gondii*, adapted from a previous study[47]. **b**, Representative cryo-EM image showing well-separated CFs following sonication and protease treatment. Overlapping boxes (yellow rectangles) were used to extract CF particles. **c**, Selected 2D class averages of CF particles, showing different views. **d**, Cross-sectional view of the cryo-EM structure of a CF segment, with component proteins displayed in distinct colors. Three clusters of proteins bound to the external tubulin surfaces are indicated by black dashed circles. **e**, Longitudinal view of the CF segment shown in **d**. **f**, Interprotofilament angles of the CF. **g**, Single-particle cryo-EM structure of CF (dark gray) fitted into a previously reported subtomogram average of CF (light orange) (EMD-66190)[34]. The red arrow indicates additional densities above the CPH1–DCX–CF6 network that are absent in our single-particle cryo-EM structure.

and cellular fractionation combined with proteomics, including the hyperplexed localization of organelle proteins by isotope tagging (hyperLOPIT) approach[27]. Collectively, these studies localize over 40 proteins to the conoid with high confidence[28], although they typically lack resolution to assign proteins to specific complexes and the current lists are unlikely to be complete because of limitations in methodology.

Recently, cryo-electron tomography (cryo-ET) and subtomogram averaging have enabled visualization of the conoid structure of apicomplexan species with unprecedented detail[10,12,17,29–34]. However, the resolutions achieved so far, typically 8–30 Å, have been insufficient for direct protein identification, thereby limiting further functional characterization.

In this study, we applied another major cryo-electron microscopy (cryo-EM) technique, single-particle analysis (SPA), to native samples purified from *T. gondii* and determined high-resolution structures of several key conoid components, including the CFs, both ICMTs, the apical region of SPMTs and one of the PCRs (Fig. 1a). With resolutions often exceeding 3.5 Å, we confidently assigned 40 different proteins to the cryo-EM densities and built their atomic models, revealing their three-dimensional (3D) organization and interaction partners. Our structures enabled rational design of synthetic lethal pairs and identification of essential genes through phenotypic screening. Our work offers unique insights into how tubulin polymers acquire specialized functions and properties through interactions with associated proteins. Furthermore, it sheds light on the molecular mechanisms underlying conoid extrusion, a critical process for apicomplexan parasite invasion and egress (exit) from host cells.

## Results

### Structure of the CFs

To determine the structure of CFs, we used an established protocol to isolate and concentrate intact conoids from detergent treated *T. gondii* cells (Methods). After proteolytic treatment with α-chymotrypsin to disassemble the conoids into separated CFs, we vitrified the samples on cryo-EM grids and collected tens of thousands of micrographs using a Titan Krios microscope (Fig. 1b). To reconstruct the CF repeat unit using single-particle cryo-EM, we extracted consecutive CF segments along the fiber axis using overlapping boxes with an 8-nm step size (Fig. 1b and Extended Data Fig. 1) and treated each segment as a single particle. Because of their twisted geometry, we successfully captured multiple views of a CF segment (Fig. 1c), enabling us to reconstruct its 3D structure at 2.9-Å resolution (Fig. 1d).

The structure revealed a twist angle of −1.67° between consecutive α,β-tubulin heterodimers within the same protofilament. In contrast, conventional microtubules typically exhibit a twist angle of -0.1° (ref. 35). This substantial deviation explains the curved morphology of CF (Fig. 1b,c), which may only be possible in the bent C-shaped configuration. We also identified a 'microtubule seam' (ref. 36) between protofilaments 4 and 5, where α-tubulin is laterally associated with β-tubulin (Fig. 1d). The angles between protofilaments 2–3 and 3–4 are unusually large compared to the 27.7° angle typical of a 13-protofilament microtubule (Fig. 1f) and are largely responsible for the bent C shape. At these sites, the canonical tubulin lateral interfaces are preserved but further reinforced by neighboring microtubule-associated proteins (MAPs).

The high-quality side-chain densities of our structure (Extended Data Fig. 2) also allowed the identification of 12 different proteins bound to the tubulin surfaces. Six of these proteins were renamed CF1 through CF6 (Table 1 and Extended Data Fig. 3). CF2, CF3, CF4 and SPM1 (ref. 22) are MIPs that bind to the open lumen of the CF (Fig. 2a). CF2 is particularly abundant, forming eight coiled-coil heterodimers that associate with the luminal surface of protofilaments 5–9 (Fig. 2d), with its C-terminal short helices binding across the seam (Fig. 2g, red arrow). Unlike other CF proteins, CF2 exhibits 24-nm rather than 8-nm periodicity (Fig. 2d). While CF2, CF3 and CF4 may be specific to CFs, SPM1 is also present on nearly all protofilaments of *T. gondii* SPMTs[21]. Collectively, these MIPs likely contribute to scaffolding and stabilizing the distinctive bent C-shaped geometry of CFs.

We identified three protein clusters bound to the external tubulin surfaces. The first cluster consists of conoid protein hub 1 (CPH1), doublecortin (DCX) and CF6, which form a meshwork over the crests of protofilaments 5–9 (Figs. 1d and 2a). Within this region, the interprotofilament angle stays consistently around 19° (Fig. 1f), accounting for the repetitive lateral binding pattern of the CPH1–DCX–CF6 complex (Fig. 1d,e). Both CPH1 and DCX are crucial for the structural integrity of the conoid[25,37,38]. Notably, *T. gondii* DCX (*Tg*DCX) binds to the tubulin lattice in a manner distinct from its mammalian orthologs[39] (Fig. 2b,c). The cryo-ET structure of the *T. gondii* conoid[34] shows additional densities

above the CPH1–DCX–CF6 network (Fig. 1g, red arrow), indicating that these densities were lost during our sample preparation.

The second protein cluster forms a prominent bulge on protofilaments 2–3 (Figs. 1d and 2a) and comprises SAS6L, elements from CIP3 and a proline-rich protein (PRP), all of which specifically localize to the conoid[25,27,40]. Interestingly, SAS6L homodimers form a linear array aligned along the CF axis (Fig. 2e) with interfaces similar to those of its homolog, SAS-6 (Fig. 2f), known to form the ring-like structure of the centriolar cartwheel[41]. These linear arrays likely explain the filaments observed following overexpression of SAS6L in *Toxoplasma* tachyzoites[40].

The extended protein CIP3 appears to function as a molecular 'glue', wrapping around protofilaments 2–5 (Fig. 2a) and bridging across the seam (Fig. 2h, red arrow). CIP3 penetrates the tubulin wall (Fig. 2h,i, red arrows), engaging internally with the MIPs and externally with the third protein cluster: a coiled-coil homodimer of CF1 that extends across the seam from the CPH1 molecule bound to protofilament 5 (Fig. 2h). The distal end of CF1 interacts with CF5 and two more SAS6L linear arrays (Fig. 2a,d). Comparison to cryo-ET structures[31,33,34] indicates that this assembly is positioned between neighboring CFs in cells (Figs. 1g and 3a), leading us to designate it as the 'bridging complex'.

### Structure-guided genetic disruption of identified CF proteins

Despite the bridging complex apparently linking neighboring CFs, a previous genome-wide CRISPR screen[42] indicated that all four proteins (CF1, CF5, SAS6L and CIP3) are individually nonessential (Fig. 3b) and experimental deletion of the genes encoding CIP3 and SAS6L caused only mild growth defects[25,40].

To further investigate the functional relevance of the bridging complex subunits, we tagged each gene with a C-terminal mini auxin-inducible degron (mAID) cassette (for controlled protein degradation) and a hemagglutinin (HA) epitope (for indirect immunofluorescence assay (IFA)) using CRISPR–Cas9 gene editing techniques, as previously described[25,43–45]. All four proteins localized to the CFs, as confirmed by ultrastructure expansion microscopy (U-ExM) (Fig. 3c). When cultured in the presence of auxin (indole acetic acid, IAA), CF1 and SAS6L were degraded efficiently within 1 h (Extended Data Fig. 4a,b), while CIP3 and CF5 required over 24 h for degradation. We then tested the fitness of parasites after degradation of individual proteins using a growth assay where repeated cycles of parasite invasion, replication and egress (exit) form a visible plaque on monolayers of human foreskin fibroblasts (HFFs). Conditional knockdown (cKD) induced by auxin treatment (+IAA) or complete knockout (KO) of CF1, CF5 and SAS6L individually did not cause detectable defects in plaque formation on host cell monolayers (Fig. 3d,f and Extended Data Fig. 4c), consistent with their modest fitness scores (Fig. 3b). In contrast, depletion of CIP3 led to a decrease in plaque numbers and size (Fig. 3d,f and Extended Data Fig. 4c). Notably, despite the presence of three SAS6L arrays within the CF structure (Figs. 2d and 3a), depletion of SAS6L did not alter the expression or localization of the other three proteins (Extended Data Fig. 4d,e).

Although depletion of individual bridging complex proteins did not cause severe growth defects, this outcome may be attributable to their functional overlap, despite their differences in structure, localization and interaction partners. Such functional overlap likely serves as a mechanism to enhance resilience during the extension and retraction of the conoid, processes that exert notable mechanical stress on the CFs. To investigate this possibility, we leveraged our atomic model of the CF to design synthetic pairs of interacting proteins, for example, a KO of CF1 in the background of the SAS6L–mAID–HA cell line (Extended Data Fig. 4f). In total, we generated five synthetic pairs from the four proteins and identified two synthetic lethal pairs (Δ*cip3*/CF1–mAID and Δ*cf1*/SAS6L–mAID) and one synthetic defective pair (Δ*cip3*/SAS6L–mAID) that exhibited severe growth defects, as evidenced by impaired plaquing assays in the presence of IAA (Fig. 3e,f

**Table 1 | Proteins identified by cryo-EM in this study, organized by complex**

| Number | Protein name[a] | Gene ID TGME49_ | Number of residues | Number of copies per asymmetric unit[b] | M/S ranking[c] | Location | | Growth phenotype[d] | | References |
|---|---|---|---|---|---|---|---|---|---|---|
| | | | | | | hyperLOPIT (Markov chain Monte Carlo) | This study | CRISPR score | Experimental | |
| 1 | CF1 | 222350 | 410 | 2 | 77 | Apical 1 | CF | −1.31 | Normal | [25], this study |
| 2 | CF2 | 258090 | 954 | 16[e] | 18 | Apical 1 | CF | −1.34 | Normal | 25 |
| 3 | CF3 | 255895 | 556 | 1 | 81 | Apical 1 | CF | 0.23 | N/A | 25 |
| 4 | CF4 | 246720 | 502 | 1 | 59 | Apical 2 | CF | 0.24 | Normal | 25 |
| 5 | CF5 | 297180 | 871 | 1 | 74 | Apical 1 | CF | −1.52 | Normal | This study |
| 6 | CF6 | 245640 | 416 | 4 | 180 | Apical 1 | CF | −0.33 | N/A | 25 |
| 7 | CIP3 | 225020 | 1,010 | 1 | 16 | Apical 1 | CF | −2.78 | Modest | [25], this study |
| 8 | CPH1 | 266630 | 582 | 4 | 43 | Apical 1 | CF | −4.16 | Essential | 25 |
| 9 | DCX | 256030 | 256 | 4 | 191 | Apical 1 | CF | −5.03 | Strong | 37,38 |
| 10 | PRP | 291880 | 783 | 2 | 221 | Apical 1 | CF | 1.77 | Normal | 27 |
| 11 | SAS6L | 301420 | 263 | 6 | 218 | Apical 1 | CF | −1.62 | Modest | [40], this study |
| 12 | SPM1 | 263520 | 220 | 1[f] | 220 | Tubulin cytoskeleton | CF, ICMT, SPMT | 1.21 | Normal | 21,22,46 |
| 13 | ICMAP1 | 239300 | 1,232 | 2 | 6 | Apical 2 | ICMT | −0.74 | Normal | 10,49 |
| 14 | ICMAP2 | 224700 | 1,678 | 2 | 22 | Apical 2 | ICMT | −1.56 | Strong | 10 |
| 15 | ICMAP4 | 225340 | 2,041 | 1 | 44 | N/A | ICMT | 0.73 | N/A | This study |
| / | SPM1 | 263520 | 351 | 11[f] | 220 | Tubulin cytoskeleton | CF, ICMT, SPMT | 1.21 | Normal | 21,22,46 |
| 16 | TLAP2 | 232130 | 446 | 2 | 350 | Tubulin cytoskeleton | SPMT | −0.82 | Normal | 47 |
| 17 | TLAP3 (AC5) | 235380 | 583 | 1 | 80 | Apical 1 | ICMT, apical SPMT | 1.44 | N/A | 47 |
| 18 | TLAP4 | 201760 | 336 | 1 | 514 | Apical 1 | ICMT, apical SPMT | 0.54 | N/A | 47 |
| 19 | TrxL1 | 232410 | 220 | 11 | 346 | Tubulin cytoskeleton | ICMT, SPMT | 0.99 | Normal | 21,46 |
| 20 | TrxL2 | 225790 | 189 | 1 | 567 | Tubulin cytoskeleton | ICMT, SPMT | 1.98 | Normal | 21,46 |
| 21 | CAM1 | 246930 | 179 | 1 | 311 | N/A | PCR-P2 | 1.09 | Normal | 43 |
| 22 | CAM4 | 249240 | 149 | 1 | 771 | Cytosol | PCR-P2 | −5.28 | N/A | This study |
| 23 | CGP | 240380 | 4,956 | 1 | 97 | PM - peripheral 2 | PCR-P2 | −3.85 | Strong | 51 |
| 24 | FLM1 | 271780 | 2,777 | 1 | 116 | N/A | PCR-P2 | −2.75 | Essential | This study |
| 25 | FLM2 | 285990 | 1,192 | 1 | 413 | N/A | PCR-P2 | −2.97 | Essential | This study |
| 26 | FRM1 | 206430 | 5,009 | 2 | 89 | N/A | PCR-P2 | −3.24 | Essential | [6,13,52], this study |
| 27 | ICAP16 | 202120 | 1,322 | 4 | 72 | PM - peripheral 2 | PCR-P2 | −2.1 | Normal | 42 |
| 28 | MLC4 | 294390 | 172 | 1 | 951 | N/A | PCR-P2 | 0.38 | N/A | 56 |
| 29 | MyoL | 291020 | 2,484 | 1 | 58 | N/A | PCR-P2 | −1.83 | Strong | [55], this study |
| 30 | PCR4 | 201220 | 603 | 2 | 278 | N/A | PCR-P2 | −5.4 | Essential | 6 |
| 31 | PCR5 | 242320 | 1,073 | 2 | 96 | Apical 2 | PCR-P2 | −3.14 | Essential | 6 |
| 32 | PCR10 | 298010 | 2,322 | 1 | 41 | N/A | PCR-P2 | −2.55 | Normal | This study |
| 33 | PCR11 | 209490 | 505 | 1 | 269 | N/A | PCR-P2 | 0.49 | N/A | N/A |
| 34 | PCR12 | 219070 | 2,720 | 2 | 57 | N/A | PCR-P2 | −2.2 | N/A | N/A |
| 35 | PCR13 | 284620 | 2,333 | 1 | 70 | N/A | PCR-P2 | −1.02 | N/A | N/A |
| 36 | PCR14 | 311880 | 728 | 2 | 84 | N/A | PCR-P2 | 2.34 | N/A | N/A |
| 37 | PCR15 | 232560 | 1,822 | 1 | 276 | N/A | PCR-P2 | −0.4 | N/A | N/A |
| 38 | AKMT2 | 292170 | 1,599 | 1 | 131 | N/A | PCR-P2 | −4.83 | Essential | This study |
| 39 | SEC23 | 291680 | 791 | 2 | 210 | Nucleus - non-chromatin | PCR-P2, ER–Golgi interface | −5.46 | Normal | This study |
| 40 | SEC24 | 277000 | 1,019 | 2 | 396 | Nucleus - chromatin | PCR-P2, ER–Golgi interface | −4.58 | Essential | This study |

[a]Protein names highlighted in dark orange are given in this study. [b]For CF, ICMT and SPMT component proteins, the number of copies per asymmetric unit means the number of proteins per 8-nm repeat length. [c]M/S rankings are based on the total unique peptide count (Supplementary Table 1). [d]The CRISPR (fitness) score[42] and hyperLOPIT[27] data were obtained from ToxoDB[75]. Genes with low scores are predicted to be essential. Experimental growth phenotypes analyzed in this study are highlighted in blue. N/A, not available. [e]The periodicity for CF2 is 24 nm instead of 8 nm, with 16 copies of CF2 (8 homodimers) per 24-nm repeat length. [f]The actual periodicity of SPM1 is unclear because of the sequence similarity of its internal repeats (six Mn motifs)[21].

and Extended Data Fig. 4g). Transmission electron microscopy (TEM) further revealed that these synthetic lethal and defective pairs disrupt the conoid structure in both intracellular and extracellular parasites (Extended Data Fig. 5). These findings underscore the power of structure-guided genetic disruption in identifying functionally important proteins whose roles may otherwise be masked by functional redundancy or overlap.

## Structures of the ICMTs

The ICMTs are a pair of conventional 13-protofilament microtubules positioned within the conoid that facilitate rhoptry docking and discharge[10]. In cryo-EM images of isolated conoids, we often observed ICMT pairs either retained within the conoid (Fig. 4a) or dislodged from the conoid during sample preparation (Fig. 4b). The ICMT pairs were easily distinguishable from SPMTs by the presence of fibrous features on their surfaces (Fig. 4b, red arrow). To determine their structures, we first treated the ICMT pair as a single intact complex (Fig. 4c and Extended Data Fig. 6), yielding a 3D reconstruction at 13 Å resolution (Fig. 4d). We then refined each ICMT individually (designated ICMT-1 and ICMT-2), achieving a local resolution of ~3.4 Å, which enabled direct protein identification using side-chain information obtained from the cryo-EM density maps.

The cryo-EM structures revealed that ICMT-1 and ICMT-2 share an almost identical MIP arrangement (Fig. 4d), which is also the same as that of the apical SPMTs (Fig. 1a), as discussed below. The MIPs in these three microtubule types closely resemble those in our previously reported central SPMT structure[21] (Fig. 5f,g), with two subtle differences. Firstly, thioredoxin-like protein 1 (TrxL1) replaces its homolog TrxL2 (ref. 46) between protofilaments 12 and 13 (Figs. 4d and 5f, red arrows). Secondly, two additional MIPs, thioredoxin-like associated protein 3 (TLAP3) and TLAP4 (ref. 47), are present (Figs. 4e and 5g, red arrows). TLAP4 binds specifically at the microtubule seam, whereas TLAP3 spans laterally across 11 of the 13 protofilaments (Fig. 4e) and inserts residues into multiple taxol-binding pockets on different protofilaments (Fig. 5h). This microtubule-binding mode makes TLAP3 an arc-MIP, a class of proteins first identified in structures of *Chlamydomonas* central apparatus microtubules[48]. We validated the binding of TLAP3 and TLAP4 to ICMTs using U-ExM (Fig. 4f,g, white arrows).

In contrast to their similar MIP organizations, the two ICMTs have distinct MAPs asymmetrically distributed on their external surfaces (Fig. 4d). On ICMT-1, a long coiled-coil homodimer of ICMT-associated protein 1 (ICMAP1)[49] wraps around five protofilaments (Fig. 4d) with an 8-nm periodicity (Fig. 4h). The resolved α-helices of ICMAP1 represent only a small portion of the total protein (Extended Data Fig. 3); the unresolved portion may account for the long fibrous densities observed on one side of the ICMT pair in our micrographs (Fig. 4b, red arrow) and absent from ICMAP1-depleted parasites[10]. ICMT-2 is associated with a distinct ICMAP4 on its external surface (Fig. 4d,j), which contains several long α-helices (Extended Data Fig. 3).

At the interface between ICMT-1 and ICMT-2, we observed extensive connecting densities (Fig. 4d). Among these, we resolved a globular density associated with ICMT-1 with 8-nm periodicity and identified it as ICMAP2 (ref. 10). ICMAP2 forms a dimer and binds between protofilaments 3 and 4, with each copy interacting with a pair of helices of unknown identity (Fig. 4i, red arrows). The resolved globular domain of ICMAP2 constitutes only 10% of its total length, with the remaining portion likely contributing to the connecting densities observed between ICMT-1 and ICMT-2 (Fig. 4d). Consistent with its location, deletion of ICMAP2 results in dissociation of the two ICMTs[10].

## Structure of the apical SPMTs

Our U-ExM results (Fig. 4f,g) indicated that TLAP3/4 also localized to the apical region of the SPMTs (Fig. 1a), suggesting potential structural differences between apical and central regions of SPMTs. To investigate further, we determined the structure of the apical SPMT by manually

selecting SPMTs near the APR in our micrographs (Fig. 5a,b). The structure, resolved to 3.3-Å resolution, revealed an MIP organization nearly identical to that of the ICMTs (Fig. 5c). However, it differed in several aspects from the central SPMTs (Fig. 1a), whose cryo-EM data (Fig. 5f,g) were collected semiautomatically without specifically targeting the apical region[21]. The apical SPMT structure displays the presence of TLAP3/4 and a TrxL1 molecule at the binding site located between protofilaments 12 and 13 (Fig. 5c,d,f,g).

On the external surface of the apical SPMT near protofilaments 6 and 7, we observed a homodimer of TLAP2 (Fig. 5e). Previous studies have shown that TLAP2 coats the entire length of SPMTs, with a gap near the apical end[47]. Its absence in our previous cryo-EM structure of the central SPMT[21] (Fig. 5f) is likely because of different sample treatments. TLAP2 binds to a site directly facing the IMC, as revealed by a prior cryo-ET study[31], suggesting that TLAP2 may mediate the association between the SPMTs and the IMC.

## Structure of PCR

In the same set of cryo-EM images, we observed PCRs that were either attached or detached from the conoid (Fig. 6a). To reconstruct their 3D structure using single-particle cryo-EM, we extracted consecutive overlapping segments along the circumference of the ring (Fig. 6a) and treated each segment (containing ~3 repeating units) as an individual particle. Some of the rings appeared partially broken or twisted, likely because of the sample treatment needed to disassemble the conoid for single-particle cryo-EM. Serendipitously, these ring defects provided additional views (Fig. 6b), enabling us to obtain a 3D reconstruction at a resolution of 3.3 Å. Using the relative angle between consecutive repeating units in this structure (Fig. 6c, red dashed rectangle), we could extend our model into a complete ring comprising 46 units (Fig. 6c), although a slight misalignment was observed at the rejoining point (Fig. 6c, red arrow).

A recent cryo-ET study visualized three PCRs (P1, P2 and P3) at the apical end of the conoid, with P2 and P3 each containing 45–47 subunits per ring[33], consistent with our estimate of 46 subunits. Comparison of our structure to the PCR structures from cryo-ET confirmed that it corresponds to the P2 ring (Figs. 5c and 6a, and Extended Data Fig. 7a–c). The high-resolution cryo-EM densities (Extended Data Fig. 2) enabled us to identify 20 distinct subunits densely packed within a single repeat unit of PCR-P2 (Table 1, Fig. 6d,e and Extended Data Fig. 7d,e), with nine proteins present in two copies unrelated by C2 point-group symmetry, such as PCR4, PCR5, SEC23 and SEC24 (Fig. 6f,g). Among the identified proteins, 12 were not characterized previously in terms of localization and phenotype; six of these were renamed PCR10 through PCR15 (Table 1). The 20 proteins contain a variety of structural domains (leucine-rich, tetratricopeptide and ankyrin repeats) and functional domains (methyltransferase and cyclic nucleotide-binding), although not all features are resolved in the cryo-EM map (Extended Data Fig. 8).

The core of the P2 ring comprises two PCR4/5 heterodimers (Fig. 6e,f), corroborating recent findings that are both proteins are essential for maintaining PCR integrity[6]. PCR4 and PCR5 are homologous except that PCR5 has acquired a C-terminal GTPase-activating domain (Extended Data Fig. 8) that appears to be sterically inhibited from binding GTPases by ICAP16. The core also features conoid gliding protein (CGP) and a structurally related protein, PCR10. Both proteins contain a Clu central domain[50] and a tetratricopeptide (TPR)-like domain and make extensive interactions with PCR4/5. The central positioning of CGP and PCR4/5 agrees well with their indispensable roles in supporting gliding motility and egress and the failure of the conoid to extrude in PCR4/5-depleted parasites[6,51].

Cryo-ET structures of intact conoids have shown that the P2 and P3 rings are connected by regularly spaced linkers ~25 nm in length[33]. Our structures suggest that the top of these linkers is formed by elongated α-helical elements from the filamin-domain-containing proteins, FLM1

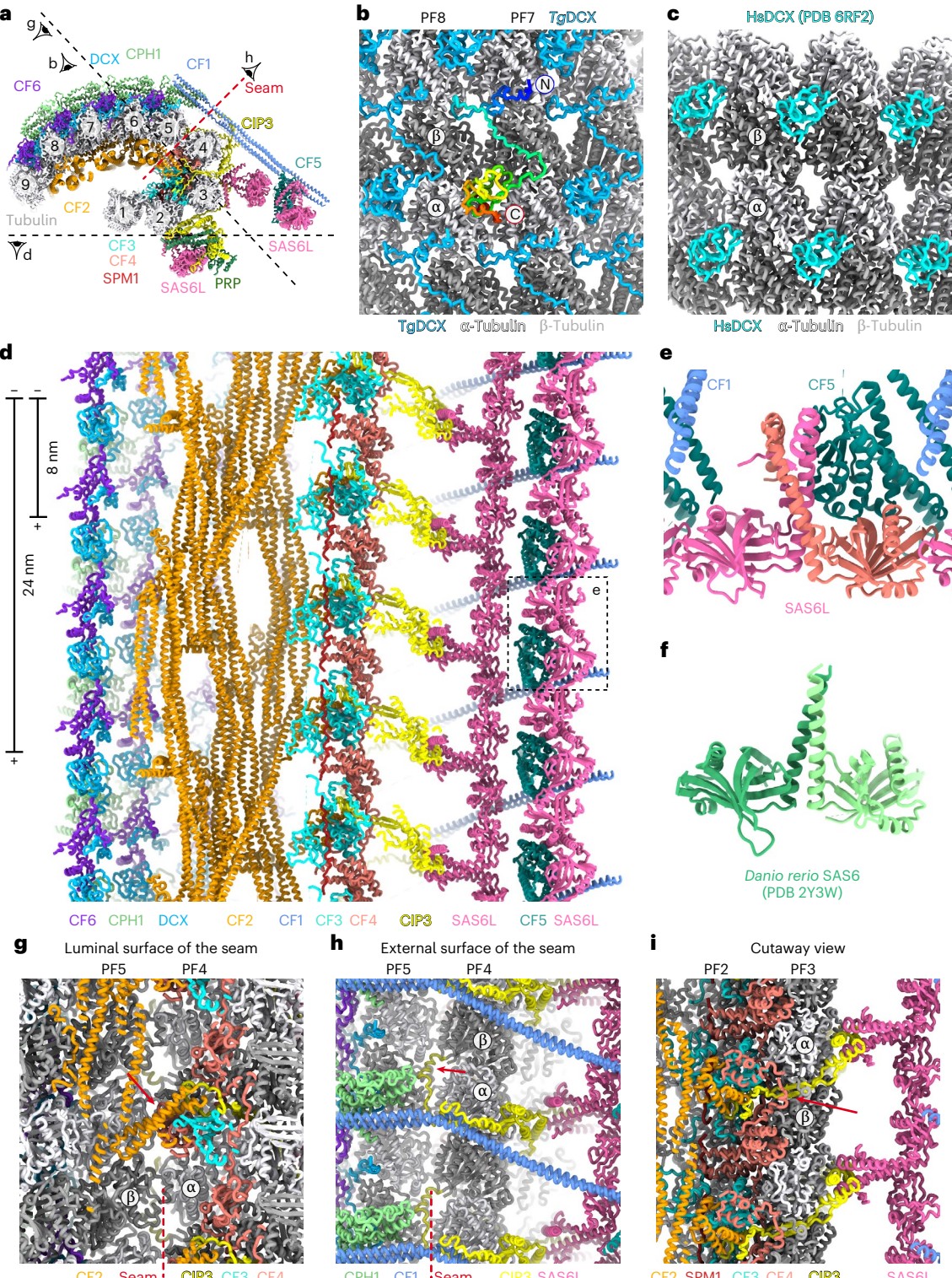

**Fig. 2 | Atomic model of the CF. a**, Cross-sectional view of the atomic model of a CF segment, with component proteins shown in distinct colors. Viewing angles for **b**, **d**, **e** and **g** are indicated. **b**, Close-up view of *Tg*DCX bound at the intradimer interface on the microtubule lattice. **c**, Close-up view of human DCX (PDB 6RF2)[39] bound at the interdimer interface on the microtubule lattice. **d**, Longitudinal view of a CF segment with 24-nm periodicity, showing component proteins in distinct colors. Atomic models of α,β-tubulin are hidden for clarity. **e**, Close-up view of *T. gondii* SAS6L linear array in the region indicated by dashed rectangles in **d**. **f**, Crystal structure of zebrafish SAS6 homodimer (PDB 2Y3W)[41]. **g**, Luminal view of the CF structure showing that C-terminal short helices of CF2 bind across the seam. **h**, External surface view of the seam between protofilaments 4 and 5. **i**, Cutaway view of the CF structure showing CIP3 penetrating through the tubulin wall.

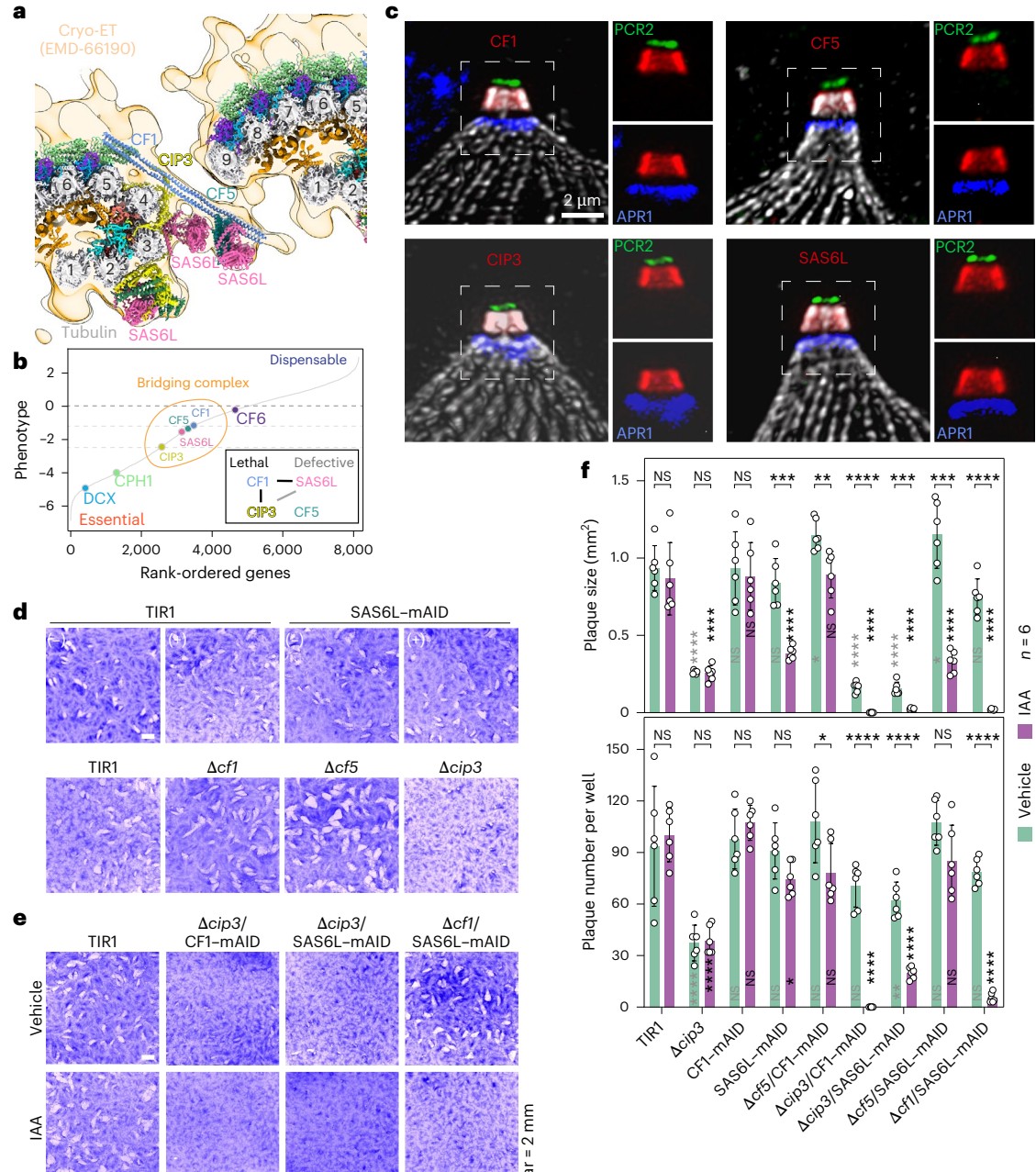

**Fig. 3 | Structure-guided genetic disruption of identified CF proteins within the bridging complex. a**, Atomic model of two adjacent CFs fitted into a previously reported subtomogram average of the CF (light orange) (EMD-66190)[34]. **b**, Positions of eight CF component genes in the phenotypic ranking of all *T. gondii* genes (*x* axis), as determined in a genome-wide KO screen[42]. The phenotype scores (fitness scores) for each gene (*y* axis) were obtained from ToxoDB[75], where genes with low scores are predicted to be essential. Inset: the two synthetic lethal pairs (black lines) and one synthetic defective pair (gray line) identified in this study. **c**, Costaining of the four bridging complex proteins with selected markers using U-ExM. Markers include PCR2 (green), APR1 (blue) and tubulin (gray). Freshly egressed extracellular parasites were labeled with chicken anti-Myc and anti-chicken IgY Alexa Fluor 405 (blue), mouse anti-Ty and anti-

mouse IgG Alexa Fluor 488 (green), rat anti-HA and anti-rat IgG Alexa Fluor 555 (red), and rabbit anti-Tubulin and anti-rabbit IgG Alexa Fluor 647 (gray). **d**, Plaque assay of four bridging complex protein mutants, generated through clean KO or cKD, on HFF monolayers treated with IAA or vehicle control (−IAA) for 8 days with 200 parasites per monolayer. Scale bar, 5 mm. **e**, Plaque assay of 2 synthetic lethal pairs and 1 synthetic defective pair on HFF monolayers treated with IAA or vehicle control (−IAA) for 8 days with 200 parasites per monolayer. Scale bar, 5 mm. **f**, Quantification of plaque area and number in parental lines and synthetic mutants (*n* = 6), from 3 independent experiments, each with 2 technical replicates. Data are shown as the mean ± s.d. Each parasite line was analyzed individually for statistical significance using an unpaired Student's *t*-test (IAA versus vehicle). NS, not significant (*P* ≥ 0.05); *$P \leq 0.05$, **$P \leq 0.01$, ***$P \leq 0.001$ and ****$P \leq 0.0001$.

and FLM2, which form a paired structure extending from the core of PCR-P2 (Figs. 5e and 6a,b). Although filamins typically function as F-actin gelation agents by crosslinking filaments, neither FLM1 nor FLM2 has the conventional actin-binding domain needed for high affinity F-actin binding. Their arrangement in the core of the P2 ring, thus, suggests that they have been co-opted for a structural role.

The surfaces of the P2 ring are functionalized by specific proteins. Notably, the outer rim contains proteins associated with actin nucleation and polymerization, including the actin-nucleator formin 1 (FRM1)[52] (Figs. 5d and 6b). The localization of FRM1 to the outer rim corresponds to the observed loss of density by cryo-ET following cKD of FRM1 in *T. gondii*[13]. Our structures demonstrate that the TPR domain of FRM1

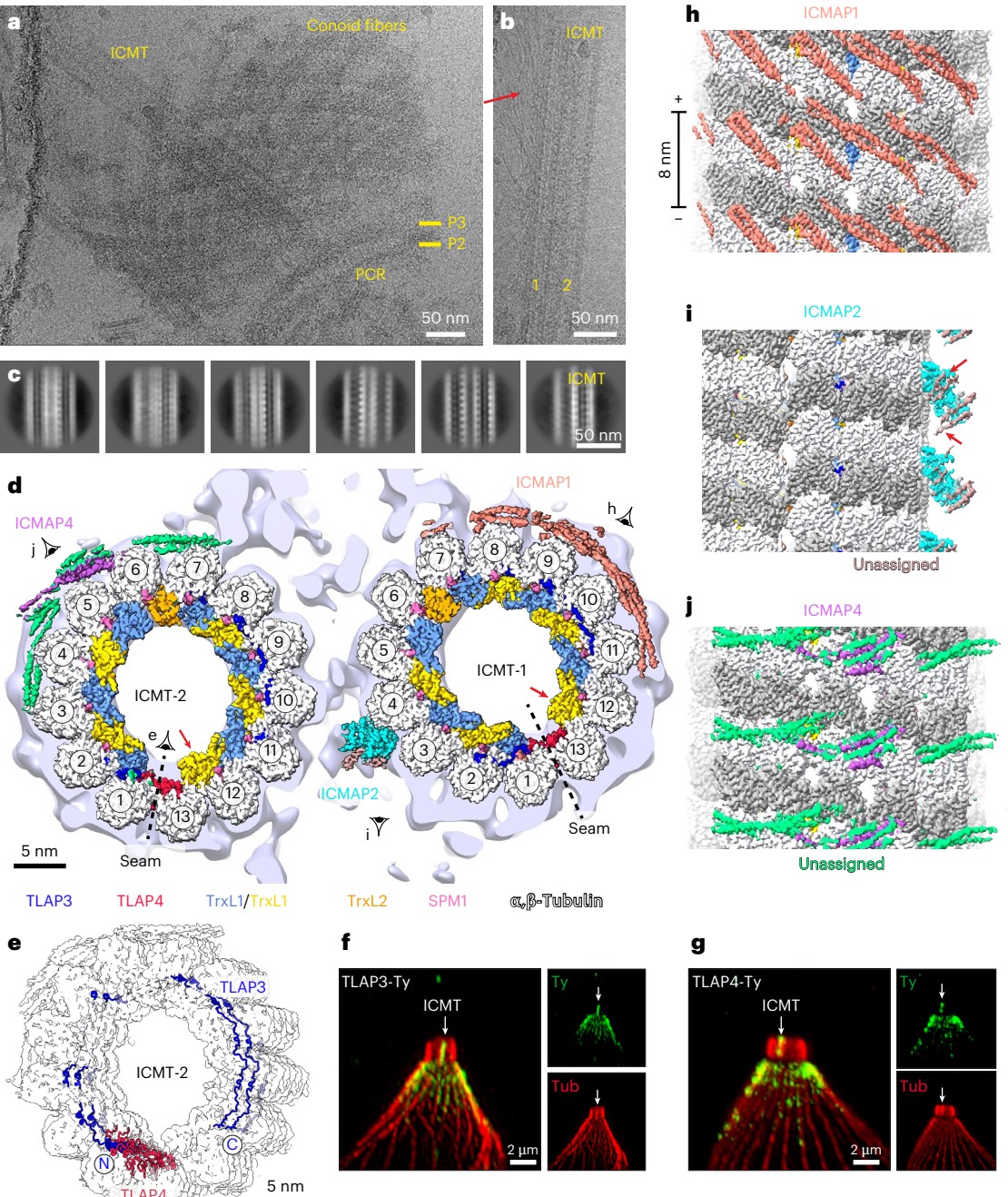

**Fig. 4 | Cryo-EM structures of ICMTs and apical SPMTs. a**, Representative cryo-EM image showing a pair of ICMTs within the conoid. **b**, Example of an ICMT pair dislodged from the conoid, revealing fibrous densities on their sides. **c**, Selected 2D class averages of ICMT particles showing different views. **d**, Cross-sectional view of the cryo-EM structures of ICMT-1 and ICMT-2, with associated proteins. The light-purple density in the background is our single-particle 3D reconstruction of the ICMT pair, obtained before refining each ICMT individually. **e**, Tilted view of ICMT-2 showing that TLAP3 spans laterally across 11 of the 13 protofilaments, while TLAP4 binds specifically at the microtubule seam. TLAP3 and TLAP4 in ICMT-1 and apical SPMT exhibit similar appearances. **f,g**, Costaining of TLAP3 (**f**) and TLAP4 (**g**) with tubulin markers using mouse anti-Ty and anti-mouse IgG Alexa Fluor 488 (green) and rabbit anti-Tubulin and anti-rabbit IgG Alexa Fluor 568 (red) for U-ExM. **h,i**, Longitudinal views of ICMT-1 showing ICMAP1 (**h**) and ICMAP2 (**i**), respectively. **j**, Longitudinal view of ICMT-2 showing MAP densities (green), with some confidently assigned to ICMAP4 (medium orchid).

embeds it within the P2 ring, while its C-terminal FH2 domain, which nucleates actin polymerization, is not resolved and may be flexible. FRM1 interacts extensively with a SET domain-containing protein, which we designate as apical lysine methyltransferase 2 (AKMT2) (Figs. 5d and 6b). This enzyme is related to, but distinct from AKMT, another lysine methyltransferase that localizes to the PCR and conoid and is required for activating parasite motility[6,53,54]. Collectively, these findings support a model where actin polymerization is initiated at the PCR by FRM1 (ref. 6).

The outer rim also contains myosin L (MyoL), a myosin motor previously localized to the conoid and cytosol[55]. We could visualize its myosin motor domain (Fig. 6d and Extended Data Fig. 7d) and its neck region interacting with three potential myosin light chain subunits (CAM1 (ref. 43), CAM4 and MLC4 (ref. 56)) (Fig. 6h). Each of these subunits contain calcium-binding EF-hand domains, suggesting that this complex may respond to calcium signaling, which is essential for conoid extrusion[25,57,58].

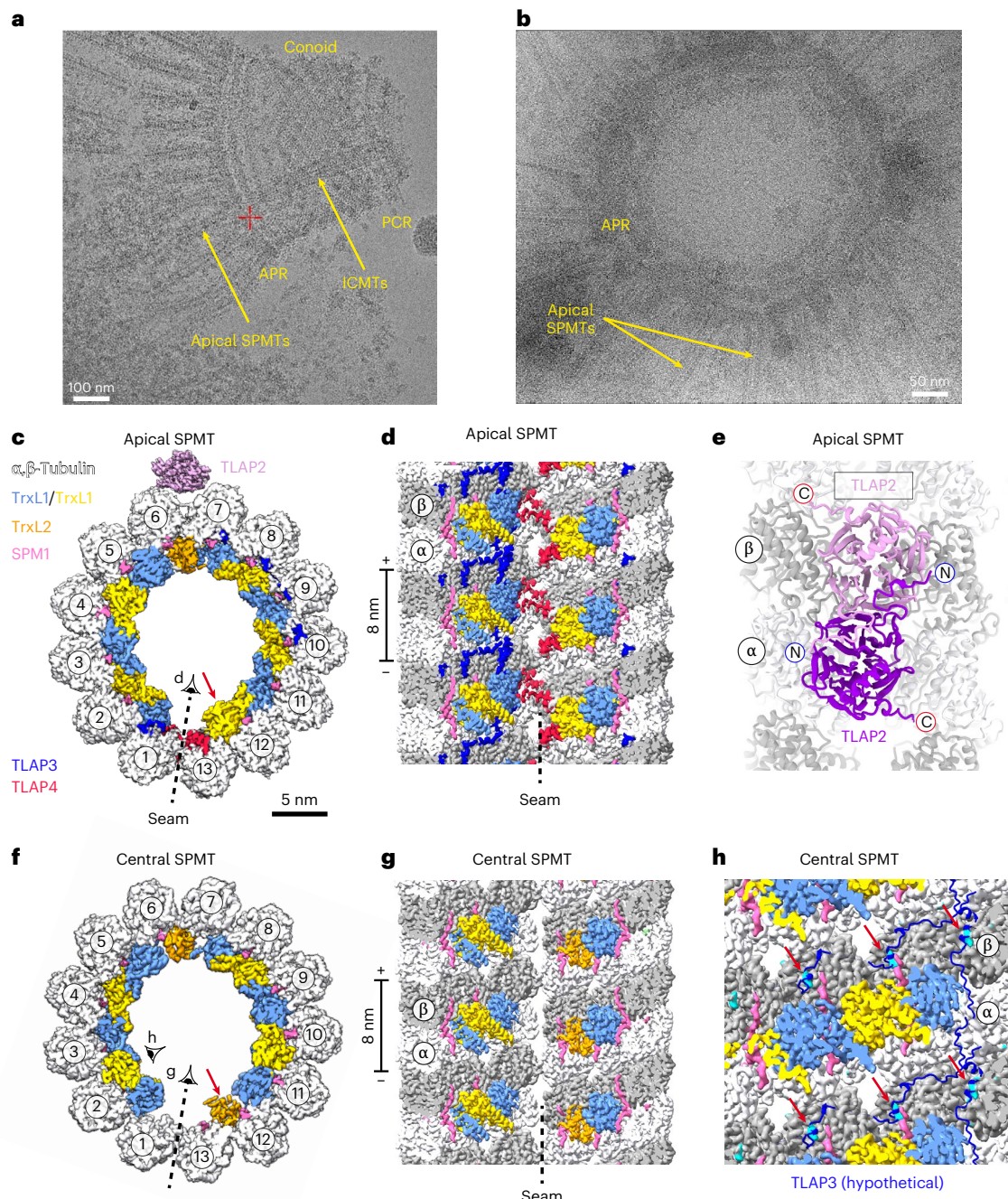

**Fig. 5 | Cryo-EM structures of apical SPMTs. a**, Representative cryo-EM image showing an isolated conoid after sonication but before protease treatment. **b**, Representative cryo-EM image of a conoid sample after sonication and protease treatment, showing SPMTs connected to the APR. **c**, Cross-sectional view of the cryo-EM structure of apical SPMT, obtained by manually selecting only the SPMTs near the APR, as shown in **a** and **b**. **d**, Luminal view of the seam in the apical SPMT. **e**, Close-up view of the TLAP2 homodimer bound to the apical SPMT. **f**, Cross-sectional view of the cryo-EM structure of the central SPMT (EMD-23869)[21]. **g**, Luminal view of the seam in the central SPMT, showing the absence of TLAP3 and TLAP4 molecules. **h**, TLAP3 inserts a segment into the taxol-binding pocket. In the central SPMT, this pocket is occupied by a small molecule-like density (cyan).

To validate our cryo-EM protein assignments and investigate their functions, we generated mAID-tagged and HA-tagged strains for eight PCR proteins: FRM1, FLM1, FLM2, AKMT2, MyoL, SEC23, SEC24 and PCR10. U-ExM confirmed their localization to the PCR (Fig. 7c and Extended Data Fig. 9d). Notably, SEC23 and SEC24, which are located at the P2 inner rim (Fig. 7b), were also detected at the endoplasmic reticulum (ER)–Golgi interface (Fig. 7e), consistent with their well-characterized functions as subunits of COPII coatamers involved in the transport of the proteins between the ER and the Golgi[59]. The dual localization of SEC23 and SEC24 implies that they maintain their protein-trafficking roles in addition to serving as structural components of the PCR.

By fusing mAID to their C termini, we achieved efficient depletion of these PCR proteins in parasites through IAA treatment, as confirmed by immunoblotting (Fig. 7c and Extended Data Fig. 9d). cKD of FLM1, FLM2, AKMT2, MyoL and SEC24 completely inhibited lytic parasite growth, as evidenced by the absence of or notable reduction in plaque formation on host cell monolayers (Fig. 7c,d), with FRM1 serving as a previously validated PCR-essential protein control[60,61]. We further demonstrated that cKD of MyoL resulted in dramatic reduction in the

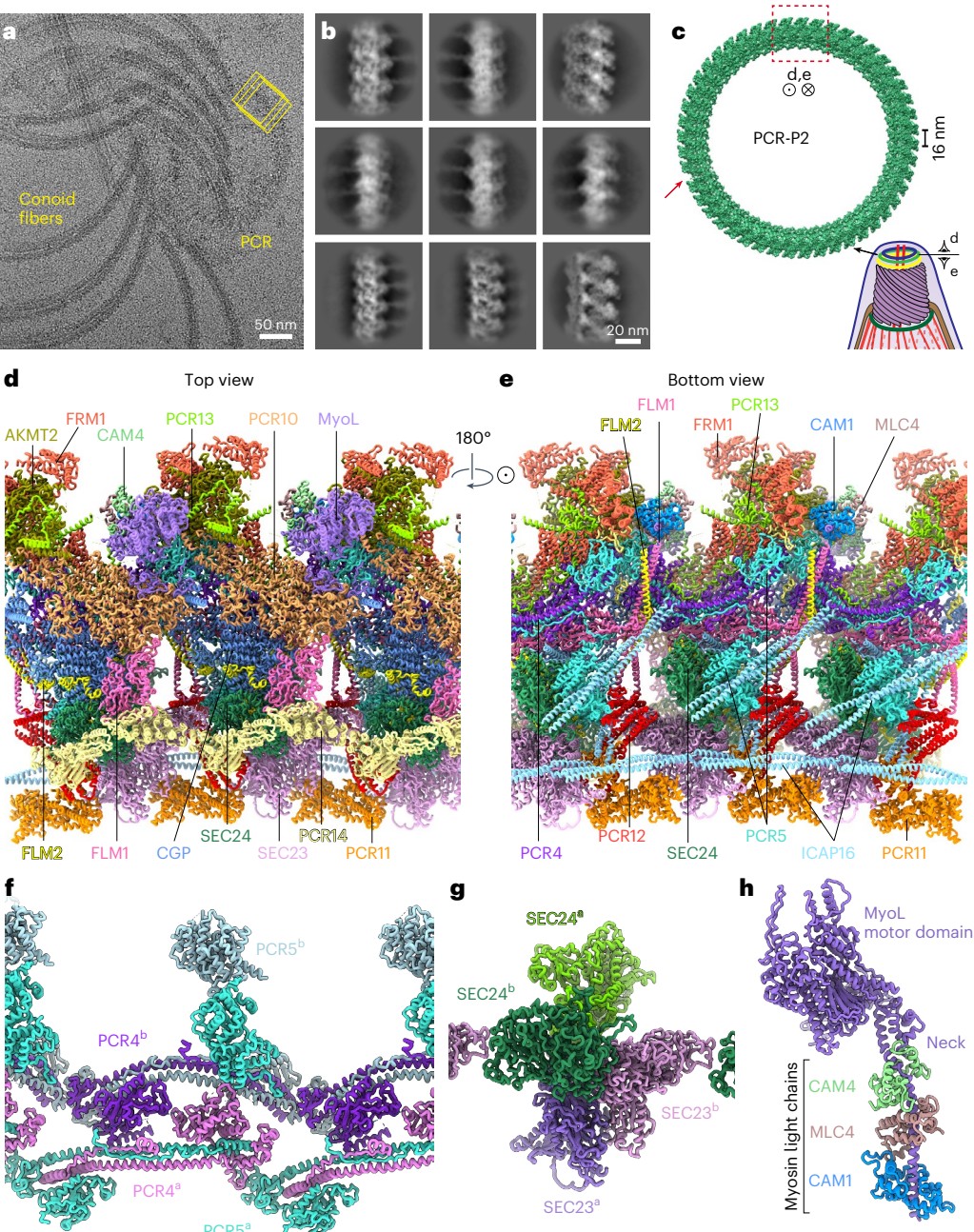

**Fig. 6 | Cryo-EM structure of PCR-P2. a,** Representative cryo-EM image of a conoid sample after sonication and protease treatment, showing a PCR dislodged from the conoid. Overlapping boxes (yellow rectangles) were used to extract the ring segments. **b,** Selected 2D class averages of ring particles showing different views. **c,** A hypothetical structure of a complete ring, generated by extending the single-particle structure (red dashed rectangle) using the measured angle between two consecutive repeating units. A slight misalignment was observed at the rejoining point (red arrow). Viewing angles for **d** and **e** are indicated. **d,e,** Top view (**d**) and bottom view (**e**) of the atomic model of the PCR-P2 containing ~3 repeating units, with component proteins shown in distinct colors. **f,** Atomic model of the two copies of PCR4/5 heterodimers located at the core of the PCR-P2. **g,** Atomic model of the two SEC23/24 heterodimers located at the inner rim of the PCR-P2. **h,** Atomic model of MyoL interacting with three potential myosin light chain subunits (CAM1, CAM4 and MLC4).

invasion assay (Extended Data Fig. 9c). In contrast, PCR10 and SEC23 are dispensable for parasite growth (Extended Data Fig. 9d), consistent with their more peripheral locations in the structure compared to their structurally related binding partners (CGP and SEC24, respectively). Thus, we identified FLM1, FLM2, AKMT2, MyoL and SEC24 as essential PCR proteins for lytic parasite growth.

## Discussion

In this study, we applied SPA cryo-EM to samples isolated from *T. gondii* and determined structures of several key conoid components, including

the CFs, both ICMTs, the apical region of SPMTs and PCR-P2. Through analysis of the cryo-EM density, we confidently assigned 40 different proteins (Table 1) and built their atomic models. Structure-guided genetic disruption of CF proteins identified functionally important proteins that might otherwise be obscured by functional overlap. Furthermore, we identified five PCR proteins (FLM1, FLM2, AKMT2, MyoL and SEC24) that are essential for the parasite's lytic growth. Our structures provide a blueprint for future functional investigations of individual conoid proteins and their collective roles. Given the structural conservation of these conoid components (Supplementary Table 1),

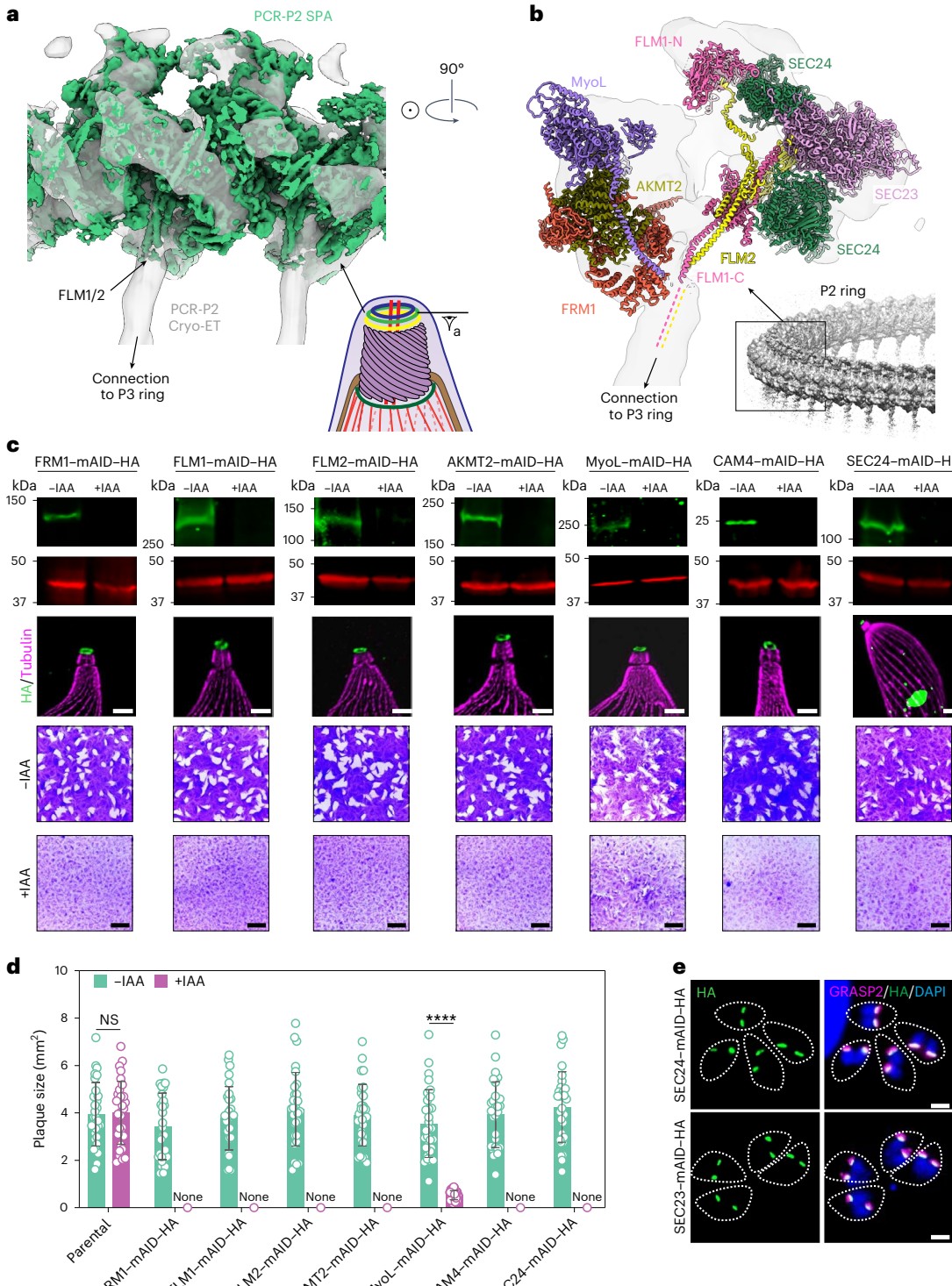

**Fig. 7 | PCR-P2 contains several essential genes. a**, The single-particle cryo-EM structure of PCR-P2 (dark green) fitted into a subtomogram average of the PCR P2–P3 rings (gray) (EMD-28126)[73]. **b**, Locations of seven identified P2 essential proteins in this study (FRM1, FLM1, FLM2, AKMT2, PCR10, CAM4 and SEC24). **c**, Rows 1–2: immunoblot of lysates from parasites containing C-terminal mAID–HA-tagged PCR proteins treated with vehicle (−IAA) or 500 μM auxin (+IAA) for 14 h. Western blots were performed for ALD1 (red) and HA (green). Row 3: localization of mAID–HA-tagged PCR proteins by U-ExM. Parasites were detected with rabbit anti-HA (green) and mouse anti-acetylated tubulin (magenta). Scale bars, 2 μm. Rows 4–5: plaque assay of mAID–HA-tagged lines on HFF monolayers treated with IAA or vehicle control (−IAA) for 8 days. Scale bars, 5 mm.

**d**, Quantification of plaque size (n = 3 independent experiments, each with n = 10 technical replicates; total n = 30). Data are presented as the mean ± s.d. Statistical analysis was conducted using a multiple-comparison unpaired Student's t-test with false discovery rate correction. ****P < 0.000001. None indicates no formed plaques. **e**, Immunofluorescence microscopy of intracellular dividing SEC23–mAID–HA and SEC24–mAID–HA parasites expressing GRASP2-Ty (Golgi marker) growing on HFF monolayers for 16 h. Parasites were detected with mouse anti-Ty (green) and rabbit anti-HA (magenta). DAPI (blue) was used to stain the nuclei of both HFFs and parasites. Scale bars, 2 μm. In these images, SEC23 and SEC24 signals at the conoid are not evident because of their notably weaker intensity compared to those at the ER–Golgi interface.

our structures also facilitate functional studies of conoid proteins in other apicomplexans, including *Cryptosporidium* and *Plasmodium* spp.

## MAPs and MIPs influence the geometry of tubulin polymers

Our structural analysis revealed the detailed organization of the tubulin protofilaments and MAPs that contribute to the unique bent C-shaped geometry of the CFs. A recent cryo-ET study suggested that CFs may originate from SPMTs[34], as some CFs in daughter cells exhibit a cylindrical shape reminiscent of conventional microtubule structures. It is intriguing to speculate that binding of the identified CF proteins during early development may guide the structural maturation of a nascent microtubule into bent C-shaped CFs. Consistent with this idea, *Tg*DCX induces strongly curved tubulin polymers when ectopically expressed in *Xenopus* cells, whereas its mammalian counterparts do not[38]. Our results may provide a structural explanation for this phenomenon. Specifically, the mammalian DCX domain binds to the tubulin lattice at the interdimer interface[39] (Fig. 2c), whereas the DCX domain of *Tg*DCX binds at the intradimer interface, with an N-terminal domain bound at the interdimer interface (Fig. 2b). These interactions likely contribute to the generation of a pronounced twist between two longitudinally interacting tubulin subunits, resulting in the curved geometry of the polymers. Our work, therefore, provides unique insight into how MAPs influence the geometry of tubulin polymers.

## MAPs impart microtubules with specialized functions

Beyond shaping tubulin geometries, MAPs also impart microtubules with specialized functions. A compelling example comes from our structural comparison of SPMTs and ICMTs, which enhance cortical rigidity[62] and facilitate repeated rounds of rhoptry discharge[10], respectively. Our structural analysis showed that both microtubule types have similar MIP organizations, presumably functioning to stabilize the microtubule. However, their MAPs differ and confer unique roles. SPMTs feature TLAP2, which we hypothesize mediates an interaction with the IMC based on its location[31], likely through the IMC component protein IMC1 (ref. 63). In contrast, the ICMTs are decorated with long coiled-coil proteins, with ICMAP1 contributing to the formation of fibrous projections from the surface of ICMT-1 (ref. 10).

## SPMT network is organized into discrete subdomains defined by MAPs and MIPs

The structural differences between apical and central SPMTs are intriguing, with TLAP3 and TLAP4 exclusively present at the apical end (Figs. 4f,g and 5d,g) and TrxL2 replacing TrxL1 on protofilaments 12 and 13 in central regions (Fig. 5c,f, red arrows). The SPMT network of *Trypanosoma* species appears to be similarly organized into discrete subdomains (posterior, middle and anterior) defined by distinct MAPs[64]. One possible explanation for these differences is that the apical and central SPMT regions experience different mechanical stresses during events such as parasite invasion and egress. This reflects an emerging concept in which associated proteins prime the microtubules with specialized mechanical properties depending on function. Supporting evidence for this idea comes from a structural comparison of mammalian axonemal doublet microtubules, which revealed that the doublet microtubules of sperm, which experience more intense forces than other doublet microtubules, exhibit the highest density of MIPs[65–68].

## The PCR serves as a platform for initiating and directing actin flow

In our structure of the PCR-P2, we find that several subunits, including PCR4, PCR5 and CGP, are deeply embedded within the ring structure and likely serve structural roles. Other essential structural subunits such as FLM1/2 and SEC24 appear to have been co-opted from other ancestral functions. These structural components create a platform for proteins involved in the processes of actin flow and conoid extrusion, which are essential for parasite invasion and egress. Previous studies proposed

that actin filament nucleation is initiated at the level of PCR and subsequently translocated by conoid-associated MyoH motors toward the APR, with the force generated by MyoH driving conoid extrusion[6,13]. Our structures expand upon this model with atomic-level details. We conclusively demonstrated that FRM1, the essential nucleator of F-actin for gliding motility, is located at the outer rim of the P2 ring (Figs. 5f and 6b), in an ideal position to initiate actin polymerization through a flexible C-terminal FH2 domain. Intriguingly, the close structural interaction with the lysine methyltransferase AKMT2 raises the possibility that actin polymerization by FRM1 may be regulated through methylation. Furthermore, our identification of MyoL within the P2 ring suggests an additional step in actin translocation before transfer to MyoH. Direct interaction of MyoL neck region with three EF-hand-containing subunits (Fig. 6h), which may serve as myosin light chains, support a role for this complex in responding to the calcium signaling that ultimately drives conoid extrusion[8]. Consistent with this idea, cKD of MyoL severely impaired parasite growth and invasion (Fig. 7c,d and Extended Data Fig. 9c). A previous study reporting that MyoL is dispensable[55] may have resulted from partial genetic disruption. Collectively, these findings reveal that PCR-P2 comprises a complex assemblage of structural components with key functional modules at the periphery.

## Structural complexity of the PCRs

The complexity of the P2 ring, with at least 20 unique proteins organized into ~46 repeating units, is similar to that of a nuclear pore complex[69] and surpasses other oligomeric ring structures including AAA+ ATPases[70], pore-forming toxin[71] and inflammasomes[72]. Dense packing of its subunits allows the formation of intricate assemblies with internal binding surfaces that stabilize the structural core, while their diverse external domains likely confer functional roles. Some domains within the structure are not resolved, such as the pleckstrin homology domains of ICAP16 and the cyclic nucleotide-binding domains of PCR12, suggesting a flexibility that may be functionally important. While many of the identified proteins were previously localized to the conoid through microscopy and proteomics, the presence of SEC23/24 at the inner rim was unexpected given their known roles in COPII coatamer formation[59]. Considering that the inner rim of the P2 ring faces the secretory organelles within the conoid, it is tempting to speculate that SEC23 and SEC24 may have a role in trafficking PCR components or secretory organelles. This interaction could have evolved into their entanglement as permanent structural components of the PCR.

Previous cryo-ET studies revealed distinct morphologies of the three PCRs[33,73]. We anticipate that each has a distinct protein composition with similar complexity to PCR-P2. Some previously identified PCR proteins not found in our P2 ring structure, such as PCR1 (ref. 15), PCR2 (ref. 26) and PCR6 (ref. 6), are likely localized to the P1 or P3 rings. In *C. parvum* and *Plasmodium falciparum*, two PCRs with similarly shaped subunits to the P2 and P3 rings in *T. gondii* were visualized by cryo-ET[13,73], suggesting that these features and their composition are conserved across the Apicomplexa phylum.

By resolving structures of conoid components, we revealed a complexity consistent with the conoid's multifaceted roles as a cytoskeleton component, a platform for actin-based gliding motility and a conduit for coordinated protein secretion. Given the complexity of the conoid, which contains proteins with many annotated functional domains, it is unlikely to have evolved de novo in a single step, nor does it seem to have been imported from a distant source, as this assemblage has no known counterpart in contemporary biology. Understanding the origin and function of the conoid would benefit greatly from studying analogous structures in related apicomplexans, as well as more distant relatives within the Myzozoan group, which span over 1 billion years of evolutionary history[74]. Insights into the structural and functional evolution of the conoid in these early-branching protozoans could also shed light on the evolution of complex macromolecular cytoskeletal structures that are conserved across eukaryotes.

## Online content

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

## Methods

### Cell culture

*T. gondii* tachyzoites were cultured in HFFs (American Type Culture Collection) at 37 °C in a 5% $CO_2$ incubator, using DMEM (Life Technologies) with a pH of 7.4, supplemented with 10% FBS (GE Healthcare Life Sciences), 10 μg ml$^{-1}$ gentamicin (Thermo Fisher Scientific) and 10 mM glutamine (Thermo Fisher Scientific) for maintenance. The e-Myco plus kit (Intron Biotechnology) was used to test all strains and host cell lines, confirming their *Mycoplasma*-free status. Parasite lines used in the study are listed in Supplementary Table 1. *T. gondii* type I strain RH TIR1–3FLAG[43,76] was used as the original strain for genetic modification.

### Plasmid construction

Plasmids were constructed by assembling DNA fragments using the NEBuilder HiFi DNA assembly kit (New England Biolabs). Plasmids used in this study are all listed in Supplementary Table 1.

### Primers

All primers were synthesized by Integrated DNA Technologies and are listed in Supplementary Table 1.

### Parasite transfection

Fresh parasites were obtained by mechanically scraping HFF monolayers, passing through 25G needles and filtration through 5-μm polycarbonate membranes. Purified parasites were washed with cytomix buffer (10 mM $KPO_4$, 120 mM KCl, 0.15 mM $CaCl_2$, 5 mM $MgCl_2$, 2 mM EDTA, 25 mM HEPES, 2 mM ATP and 5 mM GSH) and then transfected with plasmids and amplicons according to a previous protocol[77]. Briefly, $1 \times 10^7$ parasites were suspended in 300 μl of cytomix buffer and combined with a maximum of 50 μl of purified plasmid (10 μg) and/or DNA amplicons (2 μg) within a 4-mm-gap BTX cuvette and electroporated using a BTX ECM 830 electroporator (Harvard Apparatus) using the following parameters: 1,700 V, 176-ms pulse length and two pulses with a 100-ms interval between pulses. Transfected parasites were cultivated within HFFs for 24 h before drug selection, as described below.

### Generation of stable transgenic parasite lines

**C-terminal tagging.** The RH TIR1–3FLAG line was used as the parental strain for C-terminal epitope tagging. Gene-specific pSAG1:CAS9-U6 (*Toxoplasma* U6 promoter, an RNA polymerase III promoter):sgRNA (3′) plasmid was generated to target the region downstream of the translation stop codon of the gene of interest. Those sgRNA-specific plasmids were cotransfected with DNA amplicons containing mAID–HA containing a floxed HXGPRT selectable marker with 40-bp homologous arms overlapping with sequences preceding the stop codon and downstream of the protospacer-adjacent motif site. For other epitope tagging, the coding sequence of mAID–HA was replaced with that of Myc and Ty using Gibson assembly. Following transfection, parasites were selected with a combination of mycophenolic acid (25 μg ml$^{-1}$) and xanthine (50 μg ml$^{-1}$). Subsequently, parasites were cloned by limiting dilution and validated by polymerase chain reaction, IFA and western blotting.

**Generation of clean KO.** A previously described CRISPR–Cas9 system[78], which contains dual sgRNAs simultaneously targeting regions adjacent to start and stop codons, was used to generate clean KO of target genes in the RH TIR1–3FLAG parental line. Dual-sgRNA-containing plasmids for the gene of interest were generated by double enzymatic digestion of the pSAG1:CAS9-U6:sgRNA (5′) using KpnI and XhoI and integration of polymerase chain reaction-amplified U6: sgRNA (3′) from pSAG1:CAS9-U6:sgRNA (3′) by Gibson assembly. The resulting dual sgRNAs containing CRISPR plasmids were cotransfected with DNA amplicons containing floxed DHFR-TS selection marker, flanked by 40-bp homologous arms near the start and stop codons. Transfected parasites were selected in 3 μM pyrimethamine. Single clones were isolated by limiting dilution in 96-well plate containing confluent HFFs and confirmed by diagnostic polymerase chain reaction.

### Indirect IFA

Intracellular parasites grown in HFF monolayers on glass coverslips were fixed in 4% (v/v) formaldehyde in PBS for 10 min, followed by permeabilization by 0.25% (v/v) Triton X-100 diluted in PBS for 20 min. Samples were then blocked by 3% BSA in PBS for 1 h at room temperature, followed by incubation with different primary antibodies and corresponding secondary antibodies conjugated to Alexa Fluor at room temperature for 1 h. PBS containing 0.1% (v/v) Triton X-100 was used to wash the samples three times after incubation with primary and secondary antibodies. Coverslips were finally mounted onto slides using ProLong gold antifade containing DAPI (Thermo Fisher Scientific) to visualize the nuclei. Images were captured and analyzed with a ×63 oil Plan-Apochromat lens (numerical aperture (NA): 1.4) on an Axioskop2 MOT Plus wide-field fluorescence microscope (Carl Zeiss). Scale bars and linear adjustments were made to images using Axiovision LE64 software (Carl Zeiss). For confocal imaging, Images were obtained through a Zeiss ×63 (NA: 1.4) objective lens, leveraging the Airyscan super-resolution functionality on the Zeiss LSM 880 confocal microscope system. Subsequent image processing and refinement were conducted using the ZEN Black software suite.

### Western blotting

Samples were prepared in CelLytic M buffer (Sigma, C2978) containing protease inhibitor cocktail (Sigma) and Benzonase nuclease (Sigma). Subsequently, samples were boiled in 5× Laemmli buffer containing 100 mM dithiothreitol for 5 min, followed by centrifugation to remove the insoluble components. Samples were separated on polyacrylamide gels by SDS–PAGE, transferred onto a nitrocellulose membrane, blocked with 5% BSA diluted in PBS (blocking buffer) and probed with primary antibodies also diluted in blocking buffer. Membranes were washed three times with PBS containing 0.1% Tween-20 (0.1% PBS-T) and subsequently incubated with goat IRDye-conjugated secondary antibodies (LI-COR Biosciences) in blocking buffer. After three additional washes, the membranes were scanned and analyzed using a LI-COR Odyssey imaging system (LI-COR Biosciences).

### Plaque assay

Freshly isolated parasites were counted using a hemocytometer (Sigma) and 200 parasites were added to 6-well plates of confluent HFF monolayers in DMEM. Cultures were then treated with either vehicle (ethanol 1:1,000) or 500 μM IAA (Sigma) to induce the knockdown of mAID fusion proteins. The plaques were allowed to develop for 7 or 8 days, followed by fixation with 100% ethanol for 5 min and staining by 0.2% (v/v) crystal violet (Sigma) for 10 min at room temperature. Finally, the six-well plates were washed with water three times. Once dried, the plaques on the monolayers were scanned by Bio-Rad ChemiDoc MP imager and analyzed by Image J (version 1.54).

### U-ExM

U-ExM was performed as previously described[79] with minor modifications. Freshly egressed tachyzoites were allowed to settle on 12-mm round poly(D-lysine)-coated (Sigma-Aldrich) coverslips for 8 min. Cells were fixed in 0.7% formaldehyde (Sigma-Aldrich) and 1% acrylamide (AA, Sigma-Aldrich) at 37 °C for 3 h. Gelation was carried out for 1 h at 37 °C in in monomer solution (19% sodium acrylate (Sigma-Aldrich), 10% AA and 0.1% *N,N′*-methylenebis-AA (Sigma-Aldrich) in PBS) containing solution ammonium persulfate (Sigma-Aldrich) and *N,N,N′,N′*-tetramethyl ethylenediamine (Sigma-Aldrich). Gels were denatured at 95 °C for 90 min and then expanded overnight in deionized water at room temperature overnight for complete expansion. The following day, gel samples were washed twice in PBS (30 min each) to remove residual water, cut into square pieces (~1 cm × 1 cm), incubated

with primary antibodies at 37 °C for 3 h and washed with 0.1% PBS-T 3 times for 10 min each. After 3 washes with 0.1% PBS-T (10 min each), secondary antibody incubation was performed for 3 h at 37 °C, followed by 3 additional washes in 0.1% PBS-T (10 min each). Gels were then washed 3 times in 0.1% PBS-T (15 min each) and expanded overnight in deionized water at room temperature. Fully expanded gels (~0.5 cm × 0.5 cm) were mounted in aqueous medium for imaging.

## Conoid sample preparation for cryo-EM
Conoids were prepared following an established protocol[38]. *T. gondii* tachyzoites (a total of ~5 × 10^9 parasites) were isolated and treated with the calcium ionophore A23187 (Sigma, 100105). The stimulated parasites were centrifuged at 800*g* for 10 min and the supernatant was discarded. The parasites were lysed by washing with EGTA-containing demembrane buffer A (5 mM CHAPS (Sigma, C3023), 1 mM Tris–acetate pH 7.5 and 10 mM K2-EGTA), followed by three washes with EGTA-free demembrane buffer B (1 mM Tris–acetate pH 7.5 and 10 mM K2-EGTA). After each wash, the parasites were centrifuged at 4,500*g* for 5 min. The lysed parasites were resuspended in EGTA-free demembrane buffer B and sonicated on ice for 10 min, using a pulse mode of 2 s on and 2 s off. The suspension became translucent with no obvious flocks and was centrifuged at 13,200*g* for 5 min; then, the supernatant was discarded. The barely white or transparent pellets were resuspended in KH buffer (20 mM K + HEPES, pH 7.5), which contained isolated intact conoids plus APRs and proximal SPMTs (Fig. 5a). To disassemble the intact conoids into separated CFs (Fig. 1b), the conoid suspension was diluted and digest with 5–10 μg ml^−1 α-chymotrypsin at room temperature or on ice for 6 min. Digestion was stopped by 100 μM TPCK (Sigma, T4376). CFs were centrifuged at 13,200*g* for 5 min and resuspended in KH buffer.

## Negative-stain EM
Isolated conoid samples were applied to glow-discharged carbon-coated copper grids (Ted Pella). After 30 s of incubation, the grid was negatively stained with 2% uranyl acetate solution and examined using a JEOL JEM-1400 120-kV TEM instrument (JEOL) at the Washington University Center for Cellular Imaging (WUCCI). Images were recorded using an AMT XR111 high-speed 4,000 × 2,000-pixel charge-coupled device camera (AMT Imaging Direct).

## Cryo-EM sample preparation and data collection
For cryo-EM, grids were prepared using a Vitrobot Mark IV (Thermo Fisher Scientific) operated at 8 °C and 100% humidity. Then, 3–4 μl of final suspension was applied to a glow-discharged R2/1 300-mesh copper grid (Quantifoil or C-Flat), blotted with filter paper and plunge-frozen in liquid ethane held at liquid-nitrogen temperature. Frozen grids were screened in a 200-keV Glacios microscope (Thermo Fisher Scientific) at WUCCI with respect to ice thickness, sample concentration and CF intactness. Low-dose images were acquired at a nominal magnification of ×57,000, with a defocus of −10 μm.

The grids with ideal qualities were stored in liquid nitrogen and transferred to a 300-keV Titan Krios microscope (Thermo Fisher Scientific) at Case Western Reserve University (CWRU) or the Pacific Northwest Center for Cryo-EM (PNCC), equipped with a BioQuantum Energy Filter (slit width: 20 eV) (Gatan) and a K3 direct electron detector (Gatan). The data were collected using SerialEM[80] at a defocus range of −0.5 to −2.5 μm. Images were recorded at a nominal magnification of ×64,000 (calibrated pixel size: 1.34 Å) or ×81,000 (calibrated pixel size: 1.07 Å and 1.059 Å for the Krios microscopes at CWRU and PNCC, respectively). Cryo-EM data were collected over 22 sessions, with all the imaging targets manually selected.

## Cryo-EM data processing
A total of 43,480 video frames were drift-corrected and dose-weighted using patch motion correction in cryoSPARC[81]. Contrast transfer function (CTF) parameters were estimated using patch CTF estimation in cryoSPARC.

**CFs.** The data-processing workflow for CFs is illustrated in Extended Data Fig. 1. CFs were automatically picked from a total of 2,792 micrographs using filament tracer in cryoSPARC. CF particles were then extracted from drift-corrected micrographs using overlapping boxes (512-pixel box size, 2× binning) with an 82.5-Å step size, corresponding to the length of an α/β-tubulin heterodimer. Two rounds of two-dimensional (2D) classifications were conducted to discard junk particles. The remaining particles were used to generate an initial model through ab initio reconstruction, followed by homogeneous refinement, yielding a preliminary 3D reconstruction of a CF segment containing ~5 tubulin dimers in length. Subsequently, the CF particles, along with their associated alignment parameters, were exported to FREALIGN (version 9.11)[82] for further local refinement. During this step, we used customized scripts (https://github.com/rui–zhang/Microtubule) to minimize alignment errors by applying the geometric relationship among neighboring CF particles[83]. The refined particle set, featuring improved alignment parameters, was reimported into cryoSPARC, reextracted with a 512-pixel box size (no binning) and subjected to a round of local refinement, followed by local CTF refinement and an additional round of local refinement. This process yielded a 3D reconstruction at 2.9-Å resolution. In this reconstruction, the core region appeared disordered. To improve the density quality in this area, we computationally removed the tubulin signals from the raw particles using particle subtraction in cryoSPARC and performed 3D classification of the tubulin-subtracted particles in RELION (version 5.0)[84] with a cylindrical mask encompassing the core region near protofilaments 5–9. This classification yielded three major classes with approximately equal particle counts, corresponding to the three possible registers of an object with 24-nm periodicity within the reconstruction box. Particles from one of the three classes were reimported into cryoSPARC, reextracted with a 512-pixel box size (no binning) and subjected to a round of local refinement, followed by local CTF refinement and an additional round of local refinement. This process ultimately produced a 3.1-Å-resolution 3D reconstruction with 24-nm periodicity.

**ICMTs and apical SPMTs.** The data-processing workflow for ICMTs is illustrated in Extended Data Fig. 6. ICMTs were manually picked from 3,491 micrographs in RELION (version 5.0)[84], with a single trace drawn for each pair of ICMTs. Isolated ICMTs were also picked as single microtubules. We then used these traces to extract the ICMT particles with overlapping boxes (800-pixel box size, 2× binning) with an 82.5-Å step size, corresponding to the length of an α/β-tubulin heterodimer. Next, the particle coordinates were imported back into cryoSPARC, where the particles were reextracted and subjected to one round of 2D classification to remove junk particles. The remaining particles were used to generate an initial model through ab initio reconstruction, followed by homogeneous refinement, yielding a preliminary 3D reconstruction of a pair of ICMTs at 12.6-Å resolution.

Subsequently, we shifted the reconstruction center to each of the ICMTs and reextracted the particles with a 512-pixel box size (2× binning). Following an established microtubule data-processing protocol[83,85], as previously applied to the CFs, the microtubule particles and their associated alignment parameters were exported to FREALIGN (version 9.11) for further local refinement. During this step, customized scripts (https://github.com/rui–zhang/Microtubule) were used to minimize alignment errors by leveraging the geometric relationships among neighboring microtubule particles. The refined particle set, along with improved alignment parameters, was reimported into cryoSPARC, reextracted with 400-pixel box size (no binning) and subjected to one round of local refinement, followed by local CTF refinement and an additional round of local refinement. Next, we performed 3D classification in cryoSPARC using a mask around the

ICMAP densities and retained the particles with strong ICMAP densities for further processing. The particle set was then expanded using symmetry expansion in cryoSPARC (rise: 82.5 Å, twist: 0°, helical symmetry order: 5) to add neighboring particles from the same microtubule. Duplicate particles were removed and the expanded particles were reextracted with a 400-pixel box size and subjected to one round of local refinement and local CTF refinement. This process ultimately yielded 3D reconstructions of ICMT-1 and ICMT-2 at 3.3-Å and 3.5-Å resolution, respectively.

To further improve the local resolution of the ICMT maps, focused refinements were performed in cryoSPARC using seven wedge masks (120 Å in length). These masks divide the microtubule radially into seven sections, each covering approximately two protofilaments. To generate maps for protein identification and model building, the local refined maps, at resolutions ranging from 3.3 to 3.5 Å, were sharpened using DeepEMhancer[86] or the phenix.auto_sharpen program in PHENIX[87]. The sharpened maps were multiplied by their corresponding masks, aligned to the consensus ICMT map and merged into a single composite map using the 'vop resample' and 'vop maximum' commands in Chimera.

The SPMTs near the APR (Fig. 5a,b) were manually picked from 5,783 micrographs and processed using the same approach as individual ICMTs.

**PCRs.** The PCRs were manually picked as single particles, with 16-nm spacing, from 7,676 micrographs using manual picker in cryoSPARC. The particles were then extracted from drift-corrected micrographs with a 512-pixel box size (2× binning). One round of 2D classification was performed to remove junk particles and the remaining particles were used to generate three initial models through ab initio reconstruction. One of these models corresponded to a segment of the PCR containing approximately three repeating units, while no other ring-like structure was detected at this stage. Next, all particles were refined using homogeneous refinement with the good initial model, resulting in a decent 3D reconstruction. Duplicate particles introduced during manual picking were removed and a customized script 'recenterVolume_PCR.py' (https://github.com/rui-zhang/Doublet) was used to perform symmetry expansion on the particle set. This script explicitly provided the rotational matrix between neighboring repeating units, which was measured using the 'measure rotation' command in Chimera. The total number of particles was effectively doubled after these operations. Subsequently, we reextracted the particles with a 512-pixel box size (no binning) and performed homogeneous refinement, followed by local CTF refinement and local refinement, with a mask cover the central repeating unit (160 Å in length). This iterative process ultimately yielded a 3D reconstruction of the PCR at 3.3-Å resolution. To allow data merging, PCR particles collected at different pixel sizes were rescaled to 1.34 Å per pixel using the 'extract from micrographs' tool in cryoSPARC.

### Protein identification, model building and refinement

Component proteins of CF, ICMTs, apical SPMT and PCR-P2 were identified using either sequence-based or structure-based approaches[88] (Extended Data Fig. 2). The sequence-based approach was applied to regions with well-resolved side chains. Protein segments were automatically modeled using artificial-intelligence-based software, including DeepTracer[89] and ModelAngelo[90]. The predicted amino acid sequences were used to search the entire *T. gondii* proteome using either protein basic local alignment search tool or a hidden Markov model[90]. If a unique hit was confidently identified and was present in the list of proteins from mass spectrometry (M/S) analysis (Supplementary Table 1), the model was updated, manually extended and combined using Coot[91]. All identified proteins were rigorously evaluated based on side-chain densities to ensure accuracy.

The structure-based approach was applied to regions of globular densities resolved at intermediate resolution (4–10 Å), where

secondary-structure elements (α-helices and β-sheets) are visible but side-chain information is unavailable. We first compiled an Alpha-Fold[92,93] library of the top 1,000 proteins identified in our M/S analysis (Supplementary Table 1) and split the predicted structures into individual domains using scripts from the DomainSeeker package[94]. Each domain was then automatically docked into the cryo-EM density map using ChimeraX[95], followed by quantitative assessments of the docking results[94,96]. Finally, the best-fitting models were visually inspected against the cryo-EM densities. Proteins were assigned only when the AlphaFold-predicted domain structure essentially matched the densities in the spatial arrangement of secondary structural elements. Using this approach, we identified TLAP2, ICMAP2, PCR11 and the myosin motor domain of MyoL.

Final models were assembled and refined against the final sharpened composite maps using phenix.real_space_refine[97]. During refinement, secondary structure, Ramachandran and rotamer restraints were applied with the weighting of nonbonded restraints set to 500. The quality of the refined model was assessed by MolProbity[98], with statistics reported in Extended Data Table 1.

### Interprotofilament rotation angle measurement

To measure the interprotofilament angle, we used the 'angle_between_domains' command in PyMOL (https://www.pymol.org/), as previously applied to doublet microtubules and central apparatus microtubules[48,99]. This command calculates the relative rotation and translation between one α-tubulin subunit and its adjacent α-tubulin subunit (or β-tubulin if across the seam) in the lateral direction.

### M/S analysis

The sample for M/S analysis was prepared using the same protocol as for cryo-EM. Isolated conoids were pelleted and suspended in KH buffer and sent for M/S analysis at the Proteomics and Metabolomics Facility at the University of Nebraska-Lincoln.

One tube of sample (358 µg) was redissolved in reducing SDS–PAGE sample buffer to a volume of 358 µl and heated at 95 °C for 10 min. Five lanes of samples constituting 10, 20, 30, 40 or 50 µg were loaded onto a Bolt 12% Bis–Tris plus gel (Thermo Fisher Scientific) and run with MES SDS running buffer. The gel was fixed for 1 h, washed briefly with water and stained overnight, followed by destaining. Two lanes of samples (40 + 50 µg = 90 µg of material) were selected and cut into three sections each and equivalent sections were pooled. All gel pieces were then cut up further into smaller pieces and washed with water, reduced by the addition of dithiothreitol and alkylated with iodoacetamide. They were then further washed in ammonium bicarbonate and acetonitrile. Trypsin was added and digestion was carried out overnight at 37 °C. Peptides were extracted from the gel pieces and dried down in a Speed-Vac. The digests were redissolved in 5% acetonitrile and 0.05% trifluoroacetic acid. M/S analysis was carried out using a 2-h gradient on a 0.075 mm × 250 mm C18 Waters CSH column feeding into an Orbitrap Eclipse M/S instrument run in OT-IT-HCD mode.

All tandem M/S samples were analyzed using Mascot (version 2.7.0; Matrix Science). Mascot was set up to search a common contaminants database (cRAP_20150130.fasta with 125 entries), a human UniProt database (82,685 sequences, downloaded November 20, 2023) and a *T. gondii* database with 8,322 sequences (ToxoDB-65_TgondiiME49_AnnotatedProteins_20230825.fasta) assuming the digestion enzyme trypsin. Mascot was searched with a fragment ion mass tolerance of 0.6 Da and a parent ion tolerance of 15 ppm.

Scaffold (version 5.1.2; Proteome Software) was used to validate tandem M/S-based peptide and protein identifications. Peptide identifications were accepted if they could be established at greater than 80.0% probability by the Peptide Prophet algorithm[100] with Scaffold delta mass correction. Protein identifications were accepted if they could be established at greater than 99.0% probability and contained at least two identified peptides. Protein probabilities were assigned by

the Protein Prophet algorithm[101]. The annotated M/S data are provided in Supplementary Table 1.

## Statistics and reproducibility

All experiments were independently repeated at least three times with similar results unless otherwise noted. Representative images shown in figures (including Figs. 1b, 3c, 4a,b,f,g, 5a,b, 6a and 7e, and Extended Data Figs. 1, 4d,e, 5a,b and 6) are from one of these independent experiments that yielded comparable outcomes. All data were collected and analyzed without blinding using Prism software (version 10.1.2; Graph-Pad). Parametric statistical tests were applied to data with a Gaussian distribution, while nonparametric tests were used for non-Gaussian populations. Statistical significance was defined as $P < 0.05$. Detailed statistical information, including the number of technical replicates, trials, s.d. and specific tests performed, is provided in the figure legends.

## Reporting summary

Further information on research design is available in the Nature Portfolio Reporting Summary linked to this article.

## Data availability

The UniProt and ToxoDB[75] databases were used for proteomic analysis. Cryo-EM structures were deposited to the EM Data Bank with accession codes EMD-72715 (CF, 24-nm repeat length), EMD-72716 (PCR-P2), EMD-72717 (ICMT-1), EMD-72718 (ICMT-2) and EMD-72719 (apical SPMT). Corresponding atomic models were deposited to the PDB with accession codes 9Y9Z, 9YA0, 9YA1, 9YA2 and 9YA3, respectively. The original M/S proteomics data were deposited to the ProteomeXchange Consortium through the PRIDE[102] partner repository with the dataset identifier PXD068413. Data and materials can be obtained from the corresponding authors upon request. Source data are provided with this paper.

## Code availability

Custom scripts used in this study are publicly available from GitHub (https://github.com/rui-zhang/Microtubule and https://github.com/rui-zhang/Doublet).

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

## Acknowledgements

We thank B. Summers and K. Basore at WUCCI, K. Li at CWRU, and M. de Farias at PNCC for their assistance with grid screening and data collection. We thank P. Li, Y. Chen and X. Wang for their contributions to particle picking of PCR. Proteomics analysis was conducted by M. Naldrett and S. Alvarez at the Proteomics and Metabolomics Facility of the University of Nebraska-Lincoln. L.D.S. and R.Z. are supported by National Institute of Allergy and Infectious Diseases grant R01AI179885.

## Author contributions

J.Z. and Y.F. prepared the *T. gondii* samples for cryo-EM. J.Z. and W.H. collected the cryo-EM images. J.Z. and R.Z. processed the cryo-EM

data. J.Z. identified the proteins and built their atomic models. Q.N. contributed to the particle picking and grid screening. Y.F. and P.Q. generated the cKD and KO strains of PCR and CF proteins, respectively, and performed the plaque assays, immunofluorescence microscopy and U-ExM. W.L.B. performed the conventional TEM experiments. A.B. contributed to the protein identification, structure analysis and paper writing. L.D.S. and R.Z. supervised the research and wrote the paper with input from all authors.

## Competing interests

The authors declare no competing interests.

## Additional information

**Extended data** is available for this paper at https://doi.org/10.1038/s41594-025-01728-w.

**Correspondence and requests for materials** should be addressed to L. David Sibley or Rui Zhang.

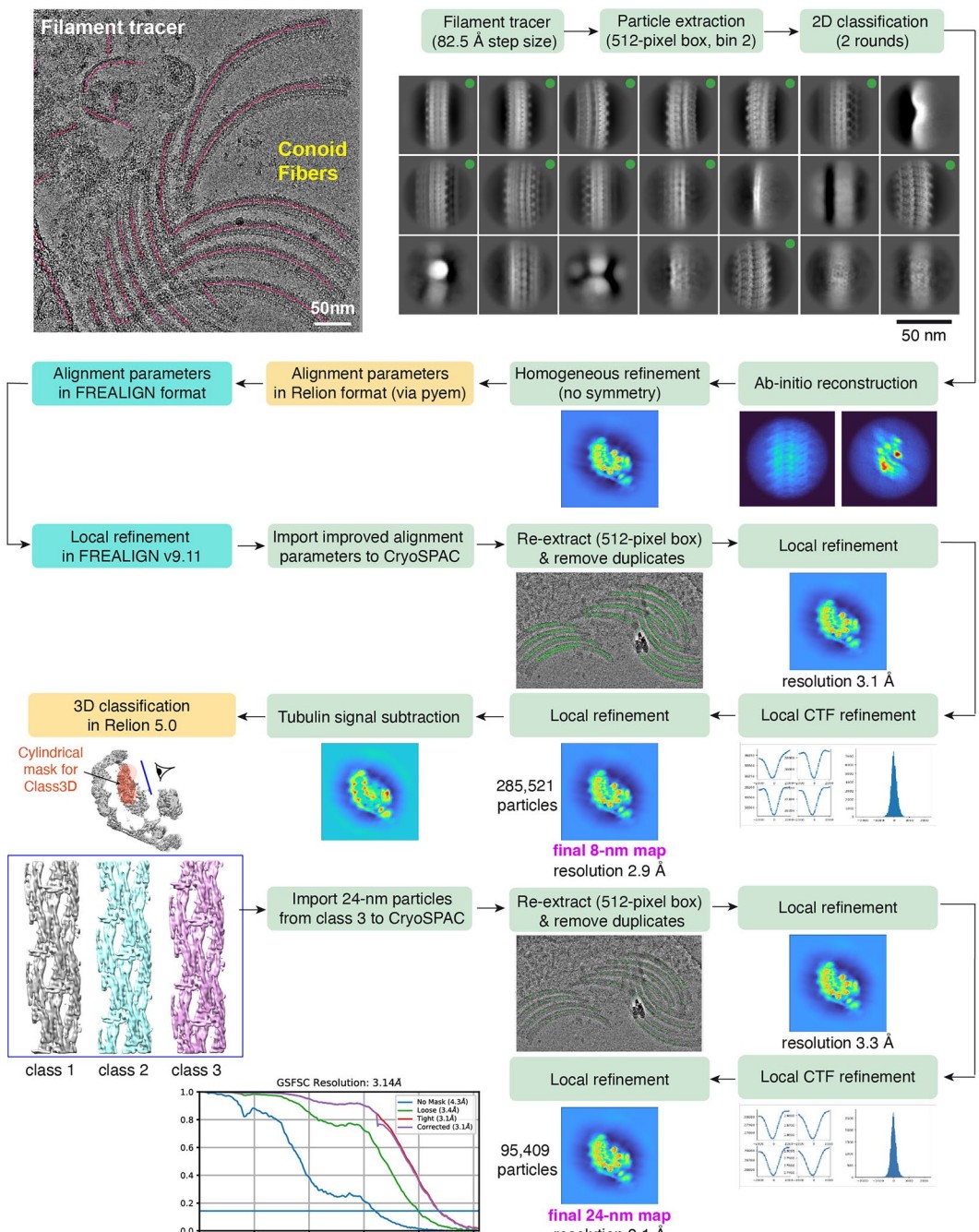

**Extended Data Fig. 1 | Data processing workflow for conoid fibers (CFs).** Operations in CryoSPARC, Relion and FREALIGN are highlighted in light green, orange and turquoise, respectively.

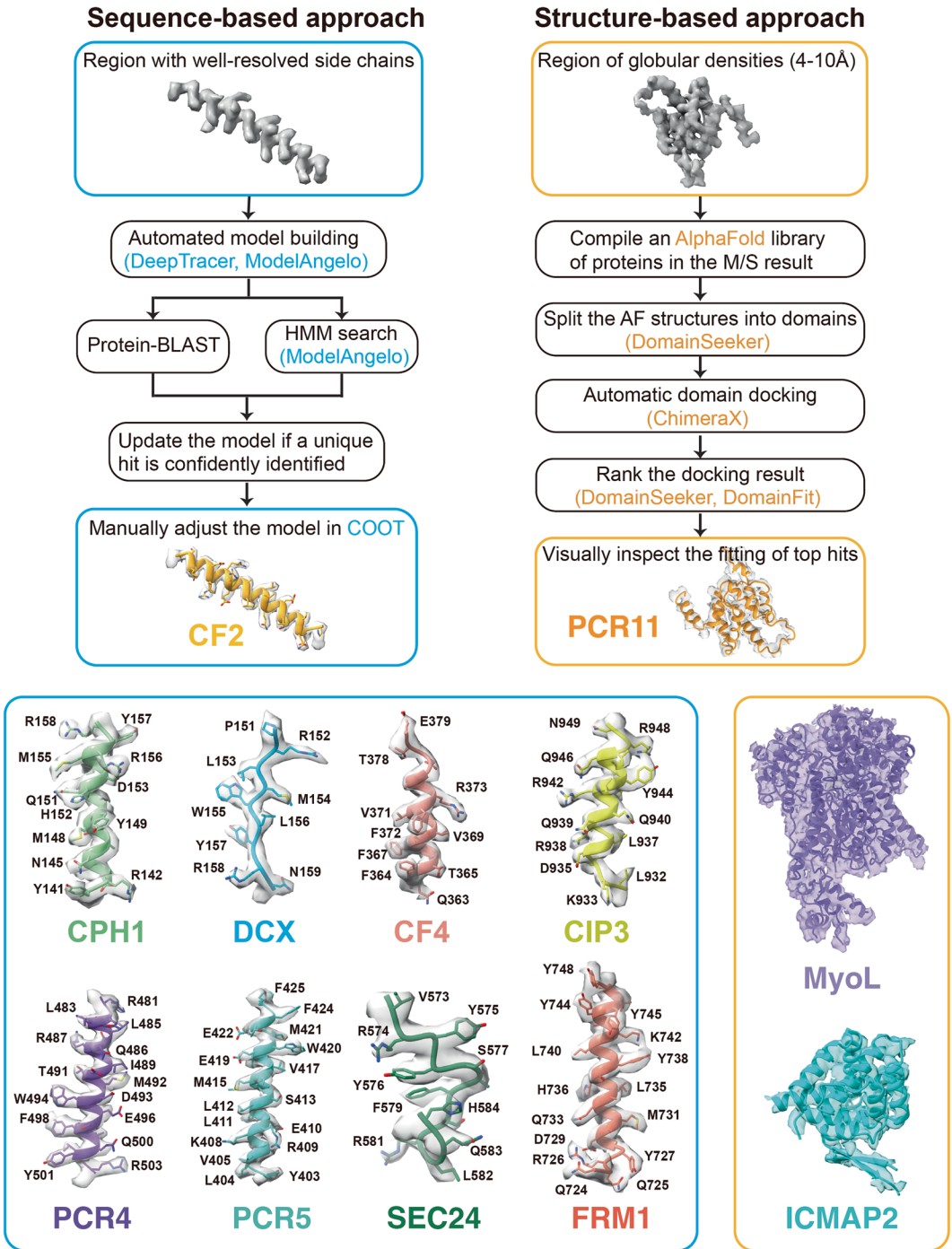

**Extended Data Fig. 2 | Protein identification workflow.** Representative examples of both sequence-based and structure-based approaches are shown.

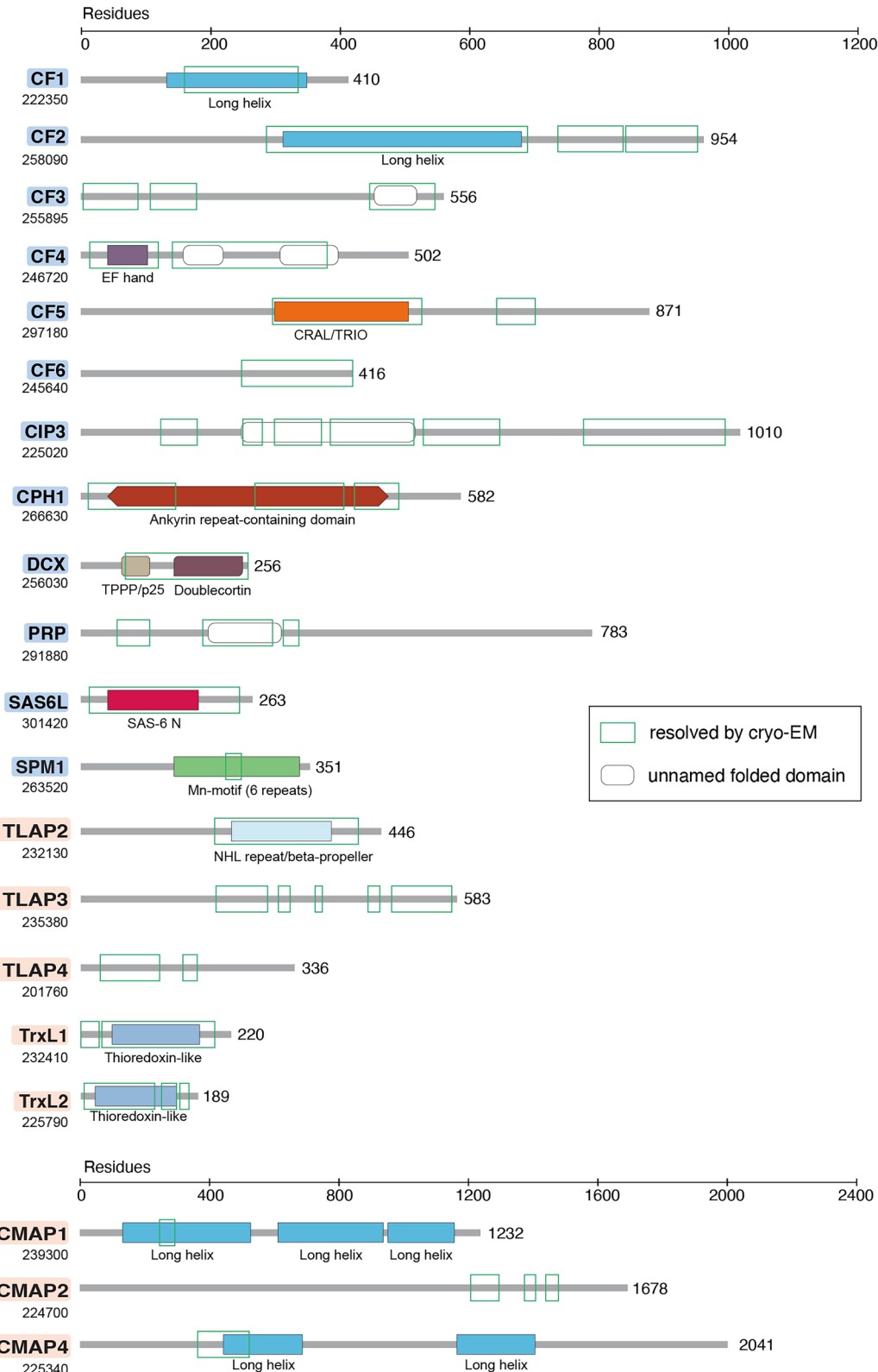

**Extended Data Fig. 3 | Domain analysis of CF and ICMT/SPMT component proteins.** Protein domain assignments are based on AlphaFold predictions[92,93], ToxoDB[75] and FoldSeek[103]. Regions resolved in the cryo-EM maps are indicated with green boxes.

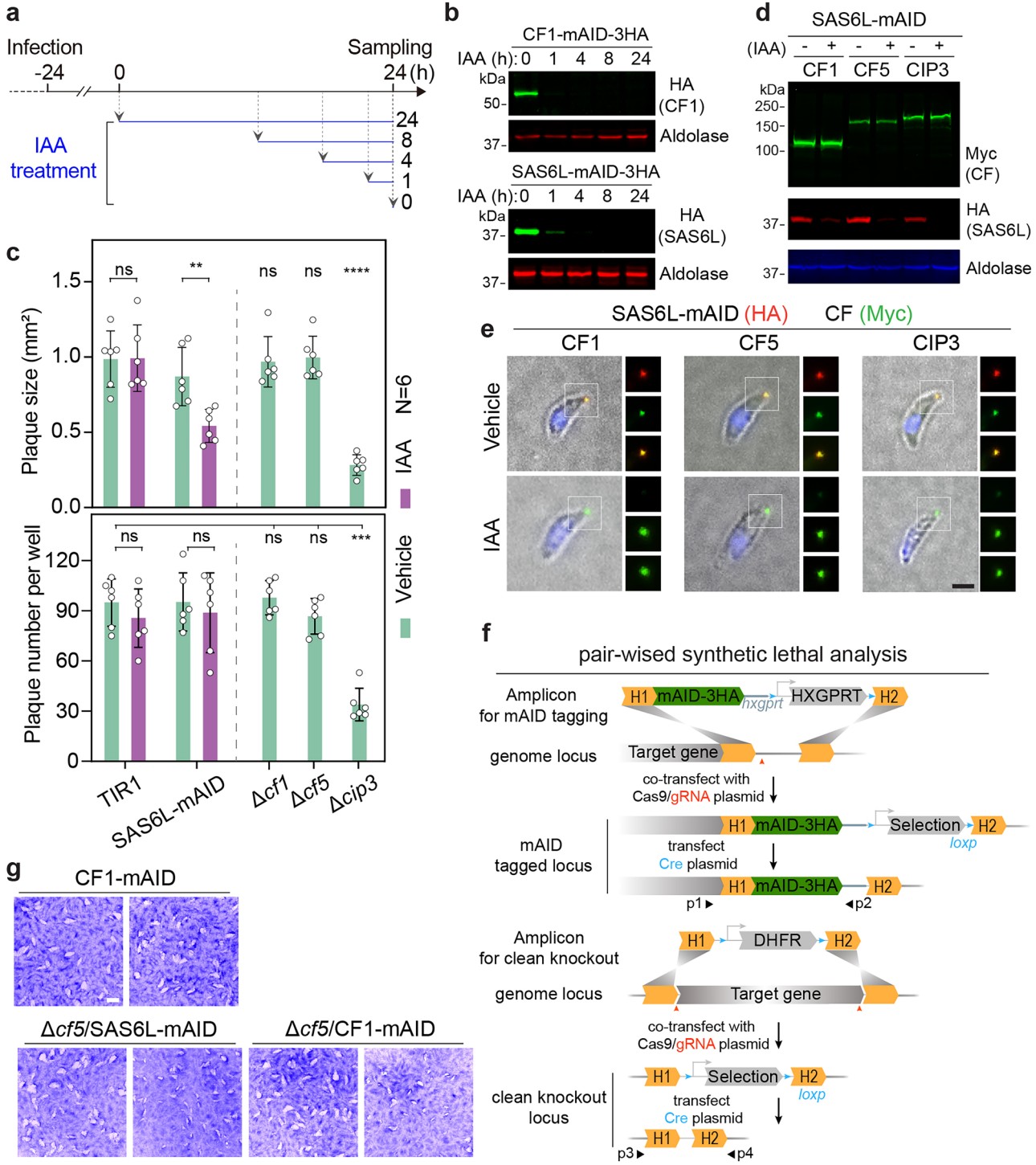

**Extended Data Fig. 4 | Synthetic lethal analysis of bridging complex proteins.**
(**a**) Time course of auxin-mediated degradation dynamics of the four bridging complex proteins. (**b**) Representative immunoblot from CF1-mAID and SAS6L-mAID lines. Blots were probed with Rabbit anti-ALD1 and anti-rabbit IgG IRDye 680RD (red), Mouse anti-HA and anti-mouse IgG IRDye 800RD (green).
(**c**) Quantification of plaque area and number in parental lines and four bridging complex protein mutants. N = 6, from 3 independent experiments, each with two technical replicates, means ± SD. Each parasite line was analyzed individually for statistical significance using an unpaired Student's t test (IAA vs. vehicle or knockout vs. TIR1), P values: ns ≥ 0.05, ** ≤ 0.01, *** ≤ 0.001, **** ≤ 0.0001.
(**d**) Immunoblot of lysates from SAS6L-mAID parasites expressing endogenously Myc-tagged CF1, CF5 and CIP3. Blots were probed with Rabbit anti-ALD1 and

anti-rabbit IgG IRDye 680RD (blue), Rabbit anti-HA and anti-rabbit IgG IRDye 680RD (red) and Mouse anti-Myc and anti-mouse IgG IRDye 800CW (green).
(**e**) Immunofluorescence (IF) microscopy of extracellular SAS6L-mAID parasites expressing endogenously Myc-tagged CF1, CF5 and CIP3. Intracellular parasites were treated with IAA or vehicle for 24 hr and freshly egressed extracellular parasites labeled with mouse anti-Myc and anti-mouse IgG Alexa Fluor 488 (green), rabbit anti-HA and anti-rabbit IgG Alexa Fluor 555 (red) and DAPI (blue). Scale bar, 1 µm. (**f**) Schematic diagram of pairwise synthetic lethal analysis used in this study. (**g**) Plaque assay of two other synthetic mutants with modest defect on HFF monolayers treated with IAA or vehicle control (-IAA) for 8 days with 200 parasites per monolayer. Scale bar, 5 mm.

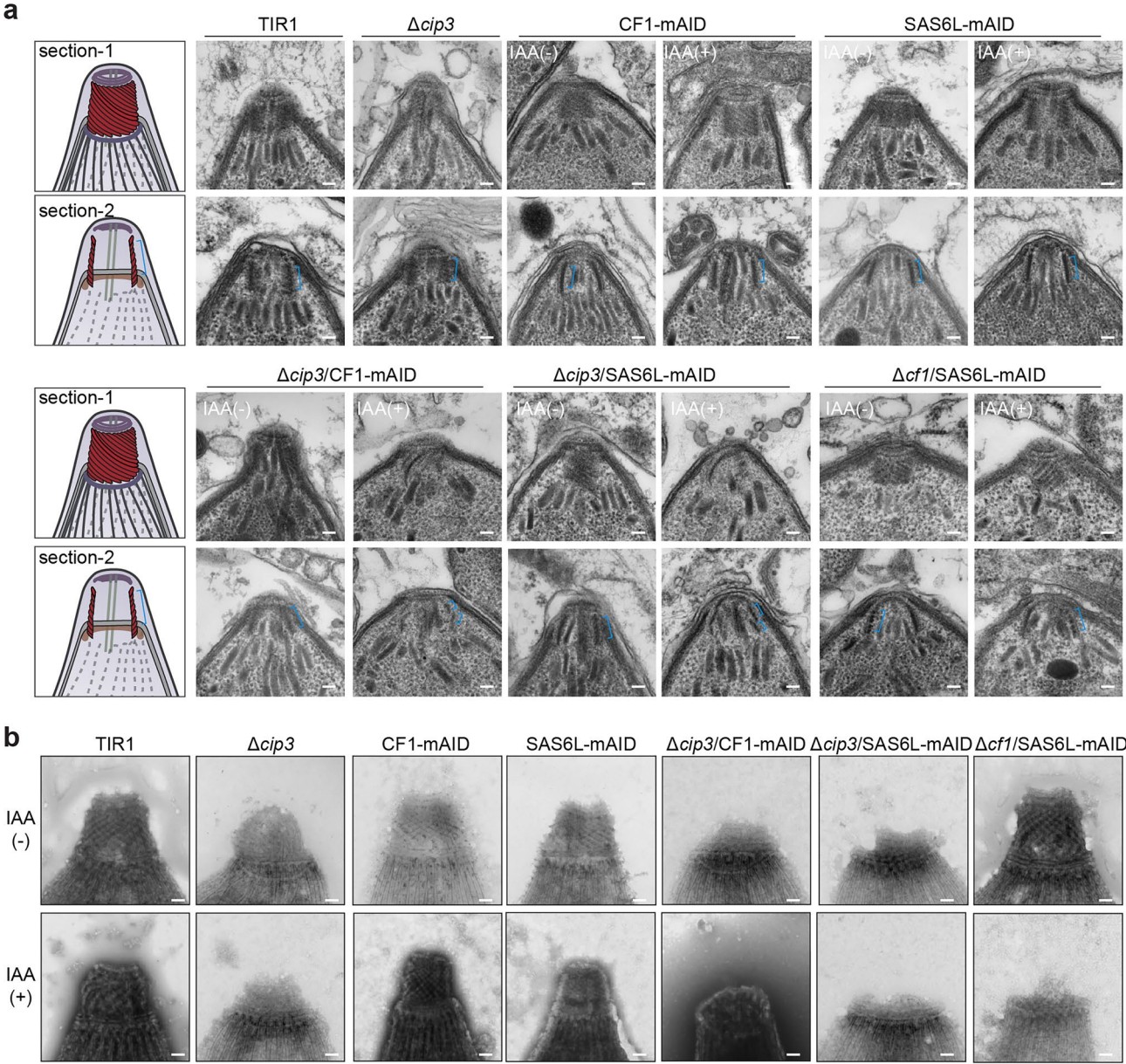

**Extended Data Fig. 5 | Synthetic lethal/defective pairs of CF proteins disrupt the conoid structure in both intracellular and extracellular parasites.**
(**a**) Transmission electron microscopy (TEM) images of intracellular parasites grown with or without IAA for 24 h. Conoid fibers are shown in tangential (section-1) and cross-sectional views (section-2). Blue brackets highlight the continuity of the conoid fibers. Scale bar = 100 nm. (**b**) TEM images of extracellular parasites prepared by negative staining. Parasites were grown with or without IAA for 24 h. Scale bar = 100 nm.

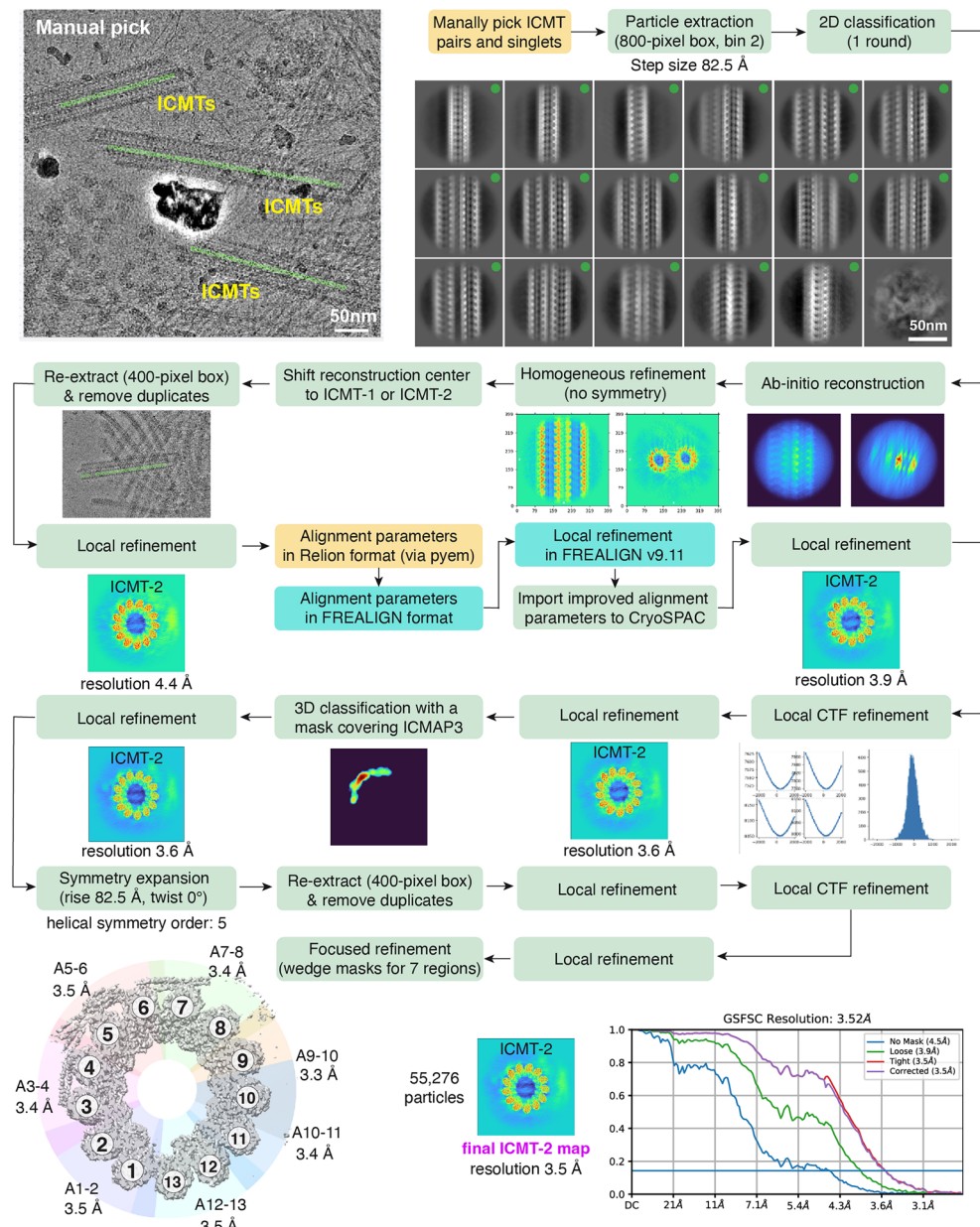

**Extended Data Fig. 6 | Data processing workflow for ICMTs.** Operations in CryoSPARC, Relion and FREALIGN are highlighted in light green, orange and turquoise, respectively.

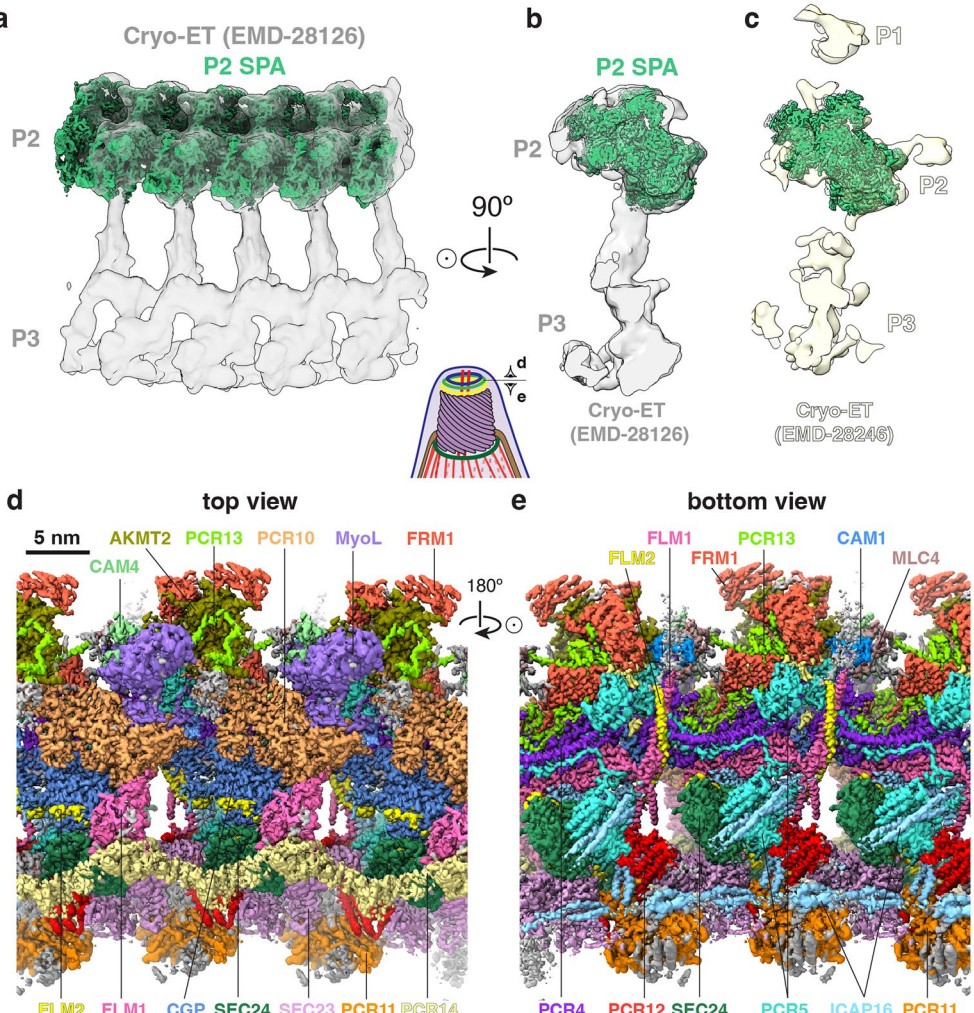

**Extended Data Fig. 7 | Cryo-EM structure of the preconoidal ring P2 (PCR-P2).** (**a, b**) Orthogonal views showing our single-particle cryo-EM structure of PCR-P2 (dark green) fitted into a subtomogram average of the PCR P2-P3 rings

(EMD-28126)[73]. (**c**) Fit of the same structure into a subtomogram averaged density of the PCR P1-P2-P3 rings (EMD-28246)[33]. (**d, e**) Top (**d**) and bottom (**e**) view of the cryo-EM density of PCR-P2 containing ~3 repeating units.

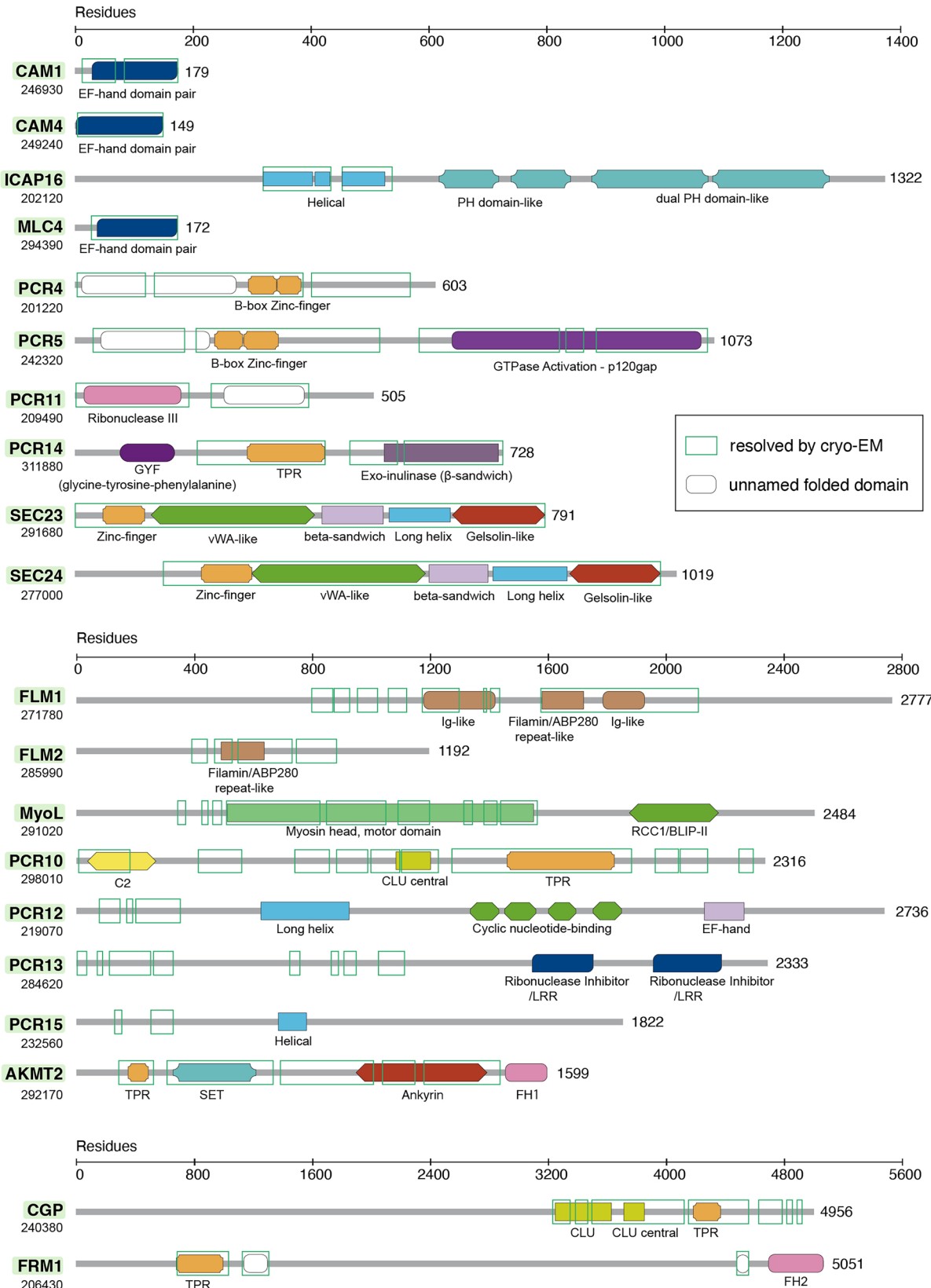

**Extended Data Fig. 8 | Domain analysis of PCR-P2 component proteins.** Protein domain assignments are based on AlphaFold predictions[92,93], ToxoDB[75] and FoldSeek[103]. Regions resolved in the cryo-EM maps are indicated by green boxes.

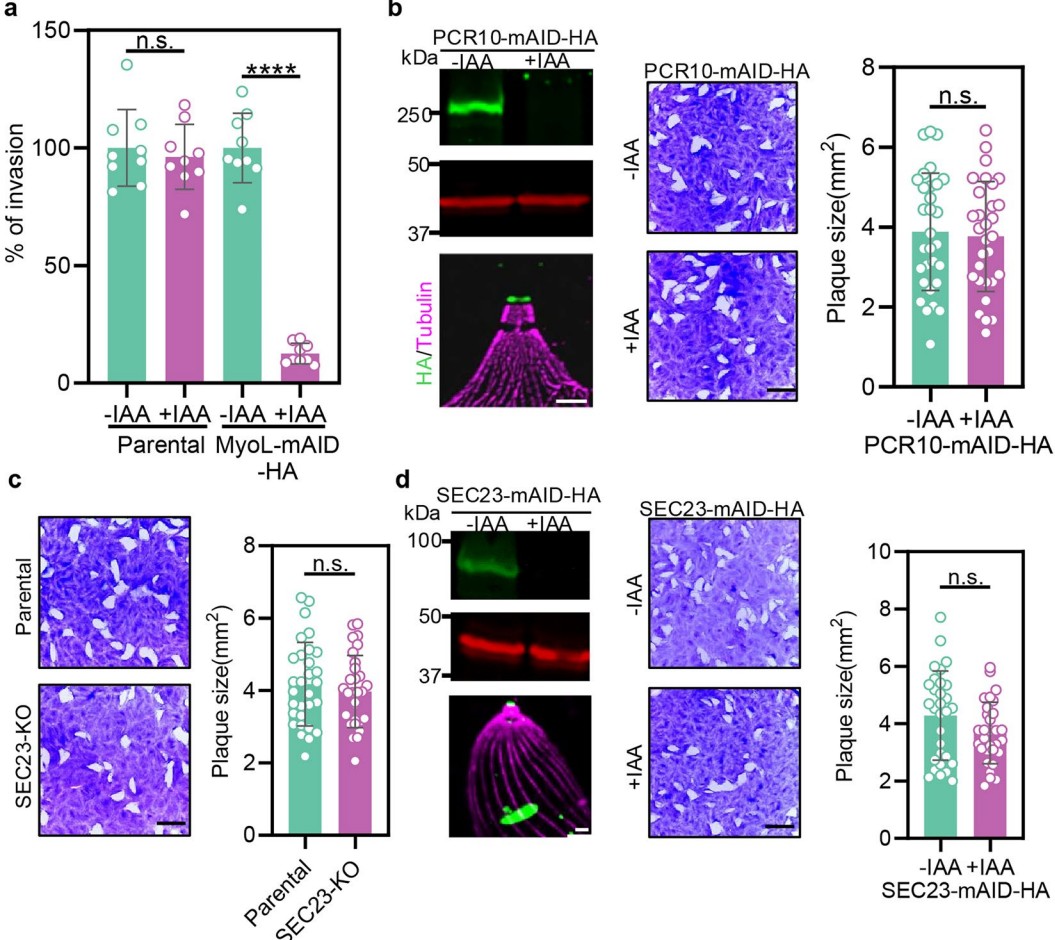

**Extended Data Fig. 9 | Functional analysis of other identified PCR proteins.**
(**a**) Invasion assay of MyoL-mAID-HA parasites. Parental and MyoL-mAID-HA parasites were pre-treated ±500 µM IAA for 24 h, then allowed to invade HFFs for 16 h before SAG1 staining. % of invasion was determined by calculating the average number of parasitophorous vacuoles per host cell and normalizing to the untreated ( − IAA) group. Means ± SD (N = 3 independent experiments, each with n = 10 technical replicates). Multiple t-test with FDR approach, n.s., not significant, ****, P < 0.000001. (**b**) Expression, localization and growth phenotype of PCR10-mAID-HA parasites. Top left: Immunoblots of parasites treated ±IAA (14 h) were probed with anti-ALD1 (red) and anti-HA (green). Bottom left: U-ExM image showing PCR10 localization (HA, green; acetyl-Tubulin, magenta). Scale bar, 2 µm. Middle: Plaque assays were performed ±IAA for

8 days on HFF monolayers (200 parasites per well). Scale bar, 5 mm. Right: Plaque sizes were shown as means ± SD from three independent experiments with 10 replicates for each (N = 3, n = 30). Two-tailed unpaired Student's t test, n.s., not significant. (c) Plaque assay showing no growth defect of SEC23-KO parasites. Scale bar, 5 mm. Data were pooled from three independent experiments with 10 replicates for each (N = 3, n = 30), means ± SD. Two-tailed unpaired Student's t test, n.s., not significant. (**d**) Expression, localization and growth phenotype of SEC23-mAID-HA parasites. Experimental design and data analysis were performed as described in (**b**). Data were pooled from three independent experiments with 10 replicates for each (N = 3, n = 30), and represented as means ± SD. Two-tailed unpaired Student's t test, n.s., not significant.

**Extended Data Table 1 | Cryo-EM processing, refinement, and validation statistics**

| Data processing | CF | ICMT-1 | ICMT-2 | Apical SPMT | PCR-P2 |
|---|---|---|---|---|---|
| No. of images | 2,792 | 3,482 | 3,435 | 5,783 | 7,676 |
| Pixel size (Å) | 1.34 | 1.34 | 1.34 | 1.34 | 1.34* |
| Picking method | CryoSPARC filament tracer | Relion manual picking | Relion manual picking | CryoSPARC filament tracer | Manual |
| Initial Model | Ab-initio reconstruction | EMD-23869 | EMD-23869 | EMD-23869 | Ab-initio reconstruction |
| Refinement method (s) | Homogeneous refinement | Homogeneous refinement | Homogeneous refinement | Homogeneous refinement | Homogeneous refinement |
| Symmetry imposed | C1 | C1 | C1 | C1 | C1 |
| Final particle images (no.) | 687,452 | 57,592 | 55,276 | 165,758 | 242,490 |
| Map resolution (Å) | 2.9 Å (8-nm) 3.1 Å (24-nm) | 3.3 (8-nm) | 3.5 (8-nm) | 3.3 (8-nm) | 3.3 |
| **Refinement** | | | | | |
| Initial model used | AlphaFold2 predictions; de novo modeling | PDB:7MIZ; de novo modeling | PDB:7MIZ; AlphaFold2 predictions; de novo modeling | PDB:7MIZ; AlphaFold2 predictions; de novo modeling | AlphaFold2 predictions; de novo modeling |
| Model composition | | | | | |
| Protein chains | 498 | 151 | 148 | 160 | 120 |
| Non-H atoms | 806376 | 231266 | 226608 | 228257 | 415081 |
| Protein residues | 100974 | 29244 | 28687 | 28883 | 52624 |
| Correlation coefficient (CCmask) | 0.56 | 0.76 | 0.67 | 0.73 | 0.53 |
| R.M.S. deviations | | | | | |
| Bond lengths (Å) | 0.006 | 0.006 | 0.006 | 0.006 | 0.005 |
| Bond angles (°) | 1.342 | 1.360 | 1.363 | 1.350 | 1.187 |
| Validation | | | | | |
| MolProbity score | 1.86 | 1.89 | 1.82 | 1.91 | 1.95 |
| Clashscore | 7.14 | 7.61 | 7.54 | 7.66 | 7.31 |
| Rotamer outliers (%) | 0.03 | 0.06 | 0.2 | 0.05 | 0.04 |
| Ramachandran | | | | | |
| Favored (%) | 92.51 | 92.34 | 93.97 | 91.92 | 89.91 |
| Allowed (%) | 7.23 | 7.48 | 5.90 | 7.83 | 9.39 |
| Outliers (%) | 0.26 | 0.18 | 0.13 | 0.25 | 0.70 |

# Reporting Summary

## Statistics

For all statistical analyses, confirm that the following items are present in the figure legend, table legend, main text, or Methods section.

| n/a | Confirmed | |
|---|---|---|
| ☐ | ☒ | The exact sample size (*n*) for each experimental group/condition, given as a discrete number and unit of measurement |
| ☐ | ☒ | A statement on whether measurements were taken from distinct samples or whether the same sample was measured repeatedly |
| ☐ | ☒ | The statistical test(s) used AND whether they are one- or two-sided<br>*Only common tests should be described solely by name; describe more complex techniques in the Methods section.* |
| ☒ | ☐ | A description of all covariates tested |
| ☐ | ☒ | A description of any assumptions or corrections, such as tests of normality and adjustment for multiple comparisons |
| ☐ | ☒ | A full description of the statistical parameters including central tendency (e.g. means) or other basic estimates (e.g. regression coefficient) AND variation (e.g. standard deviation) or associated estimates of uncertainty (e.g. confidence intervals) |
| ☐ | ☒ | For null hypothesis testing, the test statistic (e.g. *F*, *t*, *r*) with confidence intervals, effect sizes, degrees of freedom and *P* value noted<br>*Give P values as exact values whenever suitable.* |
| ☒ | ☐ | For Bayesian analysis, information on the choice of priors and Markov chain Monte Carlo settings |
| ☒ | ☐ | For hierarchical and complex designs, identification of the appropriate level for tests and full reporting of outcomes |
| ☒ | ☐ | Estimates of effect sizes (e.g. Cohen's *d*, Pearson's *r*), indicating how they were calculated |

*Our web collection on statistics for biologists contains articles on many of the points above.*

## Software and code

Policy information about availability of computer code

| Data collection | SerialEM v4.0 |
|---|---|
| Data analysis | AxioVision Se64, Graphpad Prism, Image Studio Lit, ImageJ, SnapGene, ZenBlue; cryoSPARC v3.6.0, FREALIGN v9.11, RELION v5.0, deepEMhancer v0.1, Chimera v1.17, ChimeraX v1.8, AlphaFold2, Alphafold3, Coot v0.9.8, Phenix v1.20, ModelAngelo v0.3, DeepTracer (no version number), DeepTracerID (no version number), DomainSeeker (no version number), DomainSeeker v1.11, SITUS v3.1, Mascot v2.7.0, custom scripts available from https://github.com/rui--zhang/Microtubule and https://github.com/rui--zhang/Doublet |

For manuscripts utilizing custom algorithms or software that are central to the research but not yet described in published literature, software must be made available to editors and reviewers. We strongly encourage code deposition in a community repository (e.g. GitHub). See the Nature Portfolio guidelines for submitting code & software for further information.

## Data

Policy information about availability of data

All manuscripts must include a data availability statement. This statement should provide the following information, where applicable:
- Accession codes, unique identifiers, or web links for publicly available datasets
- A description of any restrictions on data availability
- For clinical datasets or third party data, please ensure that the statement adheres to our policy

Cryo-EM structures have been deposited to the Electron Microscopy Data Bank with accession codes EMD-72715 (conoid fiber, 24-nm repeat length), EMD-72716

(PCR-P2), EMD-72717 (ICMT-1), EMD-72718 (ICMT-2) and EMD-72719 (apical SPMT). Corresponding atomic models have been deposited in the Protein Data Bank with accession codes 9Y9Z, 9YA0, 9YA1, 9YA2 and 9YA3, respectively. The mass spectrometry proteomics data have been deposition to the ProteomeXchange Consortium via the PRIDE partner repository with the dataset identifier PXD068413 and 10.6019/PXD068413.

## Research involving human participants, their data, or biological material

Policy information about studies with [human participants or human data](). See also policy information about [sex, gender (identity/presentation), and sexual orientation]() and [race, ethnicity and racism]().

| | |
|---|---|
| Reporting on sex and gender | n/a |
| Reporting on race, ethnicity, or other socially relevant groupings | n/a |
| Population characteristics | n/a |
| Recruitment | n/a |
| Ethics oversight | n/a |

Note that full information on the approval of the study protocol must also be provided in the manuscript.

# Field-specific reporting

Please select the one below that is the best fit for your research. If you are not sure, read the appropriate sections before making your selection.

☒ Life sciences ☐ Behavioural & social sciences ☐ Ecological, evolutionary & environmental sciences

For a reference copy of the document with all sections, see [nature.com/documents/nr-reporting-summary-flat.pdf](nature.com/documents/nr-reporting-summary-flat.pdf)

# Life sciences study design

All studies must disclose on these points even when the disclosure is negative.

| | |
|---|---|
| Sample size | The number of data points collected from each sample was based on the minimum required to perform statistical comparisons (n≥3). All in vitro experiments were performed at least two independent times. No statistical methods were used to predetermine sample size. For cryo-EM processing, no methods were used to predetermine sample size. The size of the cryo-EM datasets was determined by the need to identify proteins and build an atomic model. The number of micrographs and particles are listed in the Extended Data. |
| Data exclusions | Micrographs with low resolution estimates following CTF fitting were discarded. The algorithms used for image processing may down-weigh or exclude particles as part of their refinement strategy. No other data points were excluded for the analysis. |
| Replication | All in vitro experiments were repeated at least 2 or 3 times independently with 2-3 technical replicates for each. All in vitro results were successfully replicated. Two replicates of the treated T. gondii sample were subject to independent mass spectrometry analysis, which yielded similar results. Cryo-EM maps represent an average of many thousands of individual copies of the complex of interest, collected from multiple preparations across several microscope sessions. |
| Randomization | For calculation of the Fourier Shell Correlation (FSC), cryo-EM particles were randomly split into two halves. |
| Blinding | Blinding is not necessary since there are no groups that need subjective analysis. |

# Reporting for specific materials, systems and methods

We require information from authors about some types of materials, experimental systems and methods used in many studies. Here, indicate whether each material, system or method listed is relevant to your study. If you are not sure if a list item applies to your research, read the appropriate section before selecting a response.

## Materials & experimental systems

| n/a | Involved in the study |
|-----|----------------------|
| ☐ | ☒ Antibodies |
| ☐ | ☒ Eukaryotic cell lines |
| ☒ | ☐ Palaeontology and archaeology |
| ☒ | ☐ Animals and other organisms |
| ☒ | ☐ Clinical data |
| ☒ | ☐ Dual use research of concern |
| ☒ | ☐ Plants |

## Methods

| n/a | Involved in the study |
|-----|----------------------|
| ☒ | ☐ ChIP-seq |
| ☒ | ☐ Flow cytometry |
| ☒ | ☐ MRI-based neuroimaging |

# Antibodies

| | |
|---|---|
| Antibodies used | Antibodies for immunofluorescence assay (IFA) include Mouse anti-Ty, Mouse anti-HA.11 (BioLegend, Cat#901501), rabbit anti-HA (BioLegend, Cat#71-5500), Rat anti-HA (Millipore Sigma, Cat#11867423001), Mouse anti-Myc (BioLegend, Cat#626802); Chicken anti-Myc (Thermo Fisher, Cat#A21281); Mouse anti-AceTub (Sigma, Cat#T7451), Rabbit anti-AceTub (Cell Signalling Technology, Cat#5335T) and Rabbit anti-TgAldolase (in-house). For secondary antibodies, Alexa Fluor 488 Goat anti-mouse IgG (H+L) (Thermo Fisher, Cat#A-11029); Alexa Fluor 488 Goat anti-rabbit IgG (H+L) (Thermo Fisher, Cat#A-11008);Alexa Fluor 568 Goat anti-mouse IgG (H+L) (Thermo Fisher, Cat#A-11031); Alexa Fluor 568 Goat anti-rabbit IgG (H+L) (Thermo Fisher, Cat#A-11011); Goat anti-Chicken, adsorbed, DyLight 350 (Thermo Fisher, Cat#SA5-10069); IRDye 800CW Goat anti-mouse IgG (H+L) (LI-COR Biosciences, Cat#925-32210);IRDye 800CW Goat anti-rabbit IgG (H+L) (LI-COR Biosciences, Cat#925-32211); IRDye 680RD Goat anti-mouse IgG (H+L) (LI-COR Biosciences, Cat#925-68070);IRDye 680CW Goat anti-rabbit IgG (H+L) (LI-COR Biosciences, Cat#925-68071) |
| Validation | Mouse anti-HA, rabbit anti-HA antibody, Rat anti-HA, Mouse anti-Myc, Chicken anti-Myc, Mouse anti-AceTub, Rabbit anti-AceTub were validated by the manufacturers as described in the product description. Mouse anti-Ty: in house hybridoma was originally obtained from: Bastin, P., Bagherzadeh, A., Matthews, K. R. & Gull, K. A novel epitope tag system to study protein targeting and organelle biogenesis in Trypanosoma brucei. Mol. Biochem. Parasitol. 77, (1996). It was validated in the lab by testing against protein standards bearing this epitope tag. Rabbit anti-TgAldolase: Starnes GL, Jewett TJ, Carruthers VB, Sibley LD (2006) Two separate, conserved acidic amino acid domains within the Toxoplasma gondii MIC2 cytoplasmic tail are required for parasite survival. J Biol Chem 281:30745–30754 Mouse anti-TgSAG1: Burg J.L. Perelman D. Kasper L.H. Ware P.L. Boothroyd J.C. Molecular analysis of the gene encoding the major surface antigen of Toxoplasma gondii. J. Immunol. 1988; 141: 3584-3591. |

# Eukaryotic cell lines

Policy information about cell lines and Sex and Gender in Research

| | |
|---|---|
| Cell line source(s) | Human Foreskin Fibroblasts (HFFs, ATCC, Cata#CRL-1634); all T. gondii lines were obtained from published studies or generated by this study; |
| Authentication | All T. gondii strains were validated by PCR and sequencing;HFFs from ATCC were not authenticated. |
| Mycoplasma contamination | They are checked regularly using an e-MycoPlasma kit as stated in the methods. |
| Commonly misidentified lines (See ICLAC register) | No commonly misidentified cell lines were used in the study. |

# Plants

| | |
|---|---|
| Seed stocks | n/a |
| Novel plant genotypes | n/a |
| Authentication | n/a |

