## [Peer Review File · Nature Structural & Molecular Biology]

Atomic models of the Toxoplasma cell invasion machinery

Corresponding Author: Dr Rui Zhang

A version of this paper was originally rejected for publication by Nature Structural & Molecular Biology, however that decision was reconsidered after appeal by the authors.

Version 0:

Decision Letter:

2nd Jun 2025

Dear Dr. Zhang,

Thank you for submitting your manuscript "Atomic models of the conoid, the cell invasion machinery of the human parasite *Toxoplasma gondii*". Please accept our apologies for the delay in this decision. We have now carefully evaluated the work and discussed it among the editorial team. Unfortunately, we have decided not to consider the manuscript further for publication in Nature Structural & Molecular Biology.

We can only consider a small proportion of the manuscripts submitted to our journal and are often forced to make difficult decisions. Manuscripts are evaluated editorially for their potential interest to a broad audience, the level of novel insight obtained and whether the findings represent a significant advance relative to the published literature, among other considerations.

In this case, we are interested in this area of research and appreciate the application of single particle cryo-EM to the native *Toxoplasma* conoid samples, and the high resolution reconstructions obtained for the components of the conoid. However, after discussion among the editorial staff, I am afraid that despite the structural advance presented, we are not persuaded that the level of biological advance into the conoid function and *Toxoplasma* motility is sufficient to warrant publication in Nature Structural & Molecular Biology.

Although we cannot offer to publish your manuscript, I suggest that you consider Nature Communications or EMBO Journal as a suitable venue for this work. Please note that the editorial teams of those journals will make their independent assessment of your study.

To transfer your manuscript, and select the specific journal to transfer to, please use our manuscript transfer portal. You will not have to re-supply manuscript metadata and files, unless you wish to make modifications. For more information, please see our [manuscript transfer FAQ](http://www.nature.com/authors/author_resources/transfer_manuscripts.html?WT.mc_id=EMI_NPG_1511_AUTHORTRANSF&WT.ec_id=AUTHOR) page.

I am sincerely sorry we could not be more positive on this occasion. We thank you for the opportunity to consider this work and wish you success in seeking publication elsewhere.

Sincerely,
Kat

Katarzyna Ciazynska, PhD
(she/her)
Senior Editor
Nature Structural & Molecular Biology
<https://orcid.org/0000-0002-9899-2428>

** For Springer Nature Limited general information and news for authors, see <http://npg.nature.com/authors>.

Version 1:

Decision Letter:

9th Jun 2025

Dear Dr. Zhang,

Thank you for your letter concerning your manuscript "Atomic models of the conoid, the cell invasion machinery of the human parasite *Toxoplasma gondii*". We have now had a chance to discuss the points you raised in detail, and we have decided to send your paper out to review.

Before we can send the article to the reviewers, we ask that you provide data and additional documentation. Please follow the link at the bottom of this email to submit the listed information.

We also noted that the co-authors are not listed on our manuscript tracking system. Please add them when re-submitting the manuscript files.

REQUIRED MATERIALS:

1- We require official wwPDB validation reports for newly described atomic structures, as per journal policy. We also request that authors provide cryo-EM maps, half-maps and models, as well as maps and models obtained from subtomogram averaging if it applies to the work, to help the reviewers in assessing the work. We recommend the use of figshare in our system, which allows for provision of anonymous access links for the referees (<https://www.springernature.com/gp/authors/research-data/figshare-integration>). Alternatively, please upload .zip folders directly with the submission. To ensure the ease of reviewer access to the data, please specify in the Data Availability section where the files can be found (i.e., provide a figshare link or direct the reader to the manuscript files).

2- We want to ensure that the methods and statistics reporting in our papers are of the highest quality. To that end, we ask authors to fill out a Reporting Summary that collects information on experimental design and reagents, as well as an editorial Policy Checklist, which confirms compliance with our editorial policies, including the declaration of Competing Interests. If your paper includes ChIP-seq, flow cytometry or MRI data, we ask you take special care to complete those sections of the Reporting Summary as this data will aid greatly in the review of your manuscript. These documents can be found by following the links below:

Reporting Summary:

Editorial Policy Checklist: <https://www.nature.com/documents/nr-editorial-policy-checklist.pdf>

3- In order for us to proceed with peer review, please provide accession numbers and reviewer tokens to access sequencing or proteomics datasets if any unpublished datasets are part of your study. Please add this information to your manuscript file.

4- Lastly, I would like to kindly request that you provide the code used to analyse the data to the reviewers, if newly developed (unpublished) code was used in the work. For the reviewers to evaluate the work adequately, they must be able to test the software/review the code themselves. If you have not yet provided the software, we therefore request that you provide a single compressed zip file containing the software with a readme.txt file or other user manual containing complete instructions for installing and running the software. If appropriate, please provide example data and expected output. Sufficient material should be provided for referees to directly test the performance of the software/algorithm. If the software and materials are small enough to fit in a single compressed zip file under 6MB in size, you may email this file directly to me. If the zip file is between 6 MB and 200 MB, you may upload it to our file transfer site. If necessary, a second zip file up to 200 MB in size can be used to supply the example data. Please let me know if you need to use this option and I'll send you further details. Alternatively, you can upload the code to GitHub and provide us with the link.

Please fill out and return to me the code and software submission checklist that will be made available to editors and reviewers during manuscript assessment. Please note that this form is a dynamic 'smart pdf' and must therefore be downloaded and completed in Adobe Reader, instead of opening it in a web browser.

<https://www.nature.com/documents/nr-software-policy.pdf>

Once we receive these documents and review them to ensure that all requested information is provided, we will proceed to send your paper for review. If you have questions or anticipate delays, please let me know as soon as possible.

You can use the link below to be taken directly to the site and submit your manuscript:

Link Redacted

Sincerely,
Kat

Katarzyna Ciazynska, PhD
(she/her)
Senior Editor
Nature Structural & Molecular Biology
<https://orcid.org/0000-0002-9899-2428>

Version 2:

Decision Letter:

18th Jul 2025

Dear Dr. Zhang,

Thank you again for submitting your manuscript "Atomic models of the conoid, the cell invasion machinery of the human parasite *Toxoplasma gondii*". I apologize for the delay in responding, which resulted from the difficulty in obtaining suitable referee reports. Nevertheless, we now have comments (below) from the 3 reviewers who evaluated your paper. In light of those reports, we remain interested in your study and would like to see your response to the comments of the referees, in the form of a revised manuscript.

You will see that while reviewers appreciate the results, they raise several concerns which will need to be addressed in a revision. Specifically, please provide a more in depth analysis of conservation across species, in line with reviewer #1 comments. We agree with reviewer #3 that a more in depth functional investigation would strengthen the manuscript, if performing it is feasible. In line with the report of reviewer #3, please provide a more stringent statistics on model fitting into maps.

We are sorry about the comments of reviewer #2 regarding the data availability. We do appreciate you have provided the cryo-EM data for the purpose of review. We have failed to communicate their availability with this reviewer, but have since been in touch to clarify this. The data will be available again for the reviewer during re-review. We do ask, however, that you provide the final PDB validation reports with the revised manuscript.

Please be sure to address/respond to all concerns of the referees in full in a point-by-point response and highlight all changes in the revised manuscript text file. If you have comments that are intended for editors only, please include those in a separate cover letter.

We are committed to providing a fair and constructive peer-review process. Do not hesitate to contact us if there are specific requests from the reviewers that you believe are technically impossible or unlikely to yield a meaningful outcome. Particularly if you would like to discuss the feasibility of further functional experiments suggested by reviewer #3.

We expect to see your revised manuscript within 2-3 months. If you cannot send it within this time, please contact us to discuss an extension; we would still consider your revision, provided that no similar work has been accepted for publication at NSMB or published elsewhere.

Reporting Summary:
<https://www.nature.com/documents/nr-reporting-summary.pdf>

- that unprocessed scans are clearly labelled and match the gels and western blots presented in figures. Please note that all key data shown in the main figures as cropped gels or blots should be presented in uncropped form, with molecular weight markers. While these data can be displayed in a relatively informal style, they must refer back to the relevant figures. These data should be submitted as source data with the last revision, prior to acceptance.
- that control panels for gels and western blots are appropriately described as loading on sample processing controls
- all images in the paper are checked for duplication of panels and for splicing of gel lanes.
- For any revision that includes light microscopy data, we ask our authors to please include a completed light microscopy reporting table [https://www.nature.com/documents/Light_microscopy_reporting_table.xlsx] to ensure the methods are described thoroughly. The table will be available to reviewers and ultimately published should the manuscript be accepted at the journal.

EXTENDED DATA FIGURES

Data availability: this journal strongly supports public availability of data. All data used in accepted papers should be available via a public data repository, or alternatively, as Supplementary Information. If data can only be shared on request, please explain why in your Data Availability Statement, and also in the correspondence with your editor. Please note that for some data types, deposition in a public repository is mandatory - more information on our data deposition policies and available repositories can be found below:

<https://www.nature.com/nature-research/editorial-policies/reporting-standards#availability-of-data>

We require deposition of coordinates (and, in the case of crystal structures, structure factors) into the Protein Data Bank with the designation of immediate release upon publication (HPUB). Electron microscopy-derived density maps and coordinate data must be deposited in EMDDB and released upon publication. Deposition and immediate release of NMR chemical shift assignments are highly encouraged. Deposition of deep sequencing and microarray data is mandatory, and the datasets must be released prior to or upon publication. To avoid delays in publication, dataset accession numbers must be supplied with the final accepted manuscript and appropriate release dates must be indicated at the galley proof stage.

Link Redacted

Sincerely,

Katarzyna Ciazynska, PhD
(she/her)
Senior Editor
Nature Structural & Molecular Biology
<https://orcid.org/0000-0002-9899-2428>

Referee expertise:

Referee #1: toxoplasma/microbiology/parasitology

Referee #2: cryo-EM/toxoplasma

Referee #3: Toxoplasma cytoskeleton/motility

Reviewers' Comments:

Reviewer #1 (Remarks to the Author):

This outstanding manuscript uses cryo-EM and single-particle analysis to define the fine details of several protein complexes from the *Toxoplasma* conoid, an unusual organelle that is required for parasite motility and invasion. After determining protein complexes, the authors screen interacting proteins as an elegant strategy to identify important but functionally redundant protein pairs. The project is significant, and the data are of high quality. The manuscript is clear and complete. This paper has an extraordinary amount of important information, and I only have a few corrections or clarifications to suggest.

Small correction: In the figure 1 D legend, please define A as it related to protofilaments (it could be interpreted as alpha).

Line 41-2: "This stability is mainly attributed to the binding of microtubule inner proteins (MIPs)" –add ref 33 here because it demonstrates loss of stability when SPM1 is deleted (Tran et al).

Addition: "Given the structural conservation of these conoid components, our structures also facilitate functional studies of conoid proteins in other apicomplexans, including *Cryptosporidium* and *Plasmodium* spp."

It would be valuable to add one more column to table 1. For each component what is the degree of conservation across the phylum? That is, are they in all apicomplexans, in the coccidia, in other alveolates, etc. Given differences in the appearance of the conoid in the Haemosporida, I would hypothesize that some of the conoid fiber components would be lost in *Plasmodium* etc.

Questions:

1. There are two unconventional alpha tubulins in the *Toxoplasma* genome that have an insert in the N loop (TGME49_231770) or an unusually long carboxyterminal tail and missing GAP domain residues (TGME49_231400). Can either of these be placed in the structure, particularly in areas of greater structural torsion (ie between protofilaments A3 and A4)?
2. TLAP3 is an arc-MIP which spans laterally across 11 out of the 13 protofilaments and inserts residues into multiple taxol-binding pockets located at different protofilaments...Can you expand upon how TLAP3 is similar or different from *Chlamydomonas* flagellar arc-MIPs?
3. It is surprising to me that there are so many novel proteins in these complexes, given the repurposing of SFA and SAS6L in the conoid. I am surprised to not see DIP13, a homolog of a *Chlamydomonas* flagellar apparatus component that is localized to the conoid appear in this description of key proteins. Could you comment on why this may be?

Reviewer #2 (Remarks to the Author):

One of the hallmarks of apicomplexan parasites is the conserved apical complex that is essential for motility, invasion, and effector secretion. Recent advances in cryoEM and sample preparation for light microscopy have enabled a rapid increase in our understanding of this fascinating structure. In the present paper, the authors adapted a previously published protocol for the purification of the detergent-stable components of the *Toxoplasma* apical complex cytoskeleton, including the conoid and its associated microtubules. The authors averaged the cytoskeleton fibers to produce reasonably high resolution maps that allowed them to reliably identify a number of proteins known to be associated with the complex. While the apical complex is reasonably robust to loss of single proteins, the authors' structural models enabled them to predict double-mutations that would show synthetic lethality. Taken together, this really is a tour-de-force.

My comments focus on data availability, clarifying methods and model quality, and some changes to make the manuscript more accessible to a broad audience.

Major comments:

Data availability: It is frustrating that there are no deposition identifiers for any of the data (MS, cryoEM maps, or models), suggesting that the authors have not deposited any of these datasets. Given that all repositories allow embargoes there really isn't any reason for this. It's doubly confounding given that the PDB reports they provided explicitly say "not for manuscript review". This means the quality of the models and maps cannot be reliably evaluated by a reviewer. While I trust the authors as scientists, this is a bad precedent, and I thought was against NSMB policy.

The major importance of this work is the placement of specific proteins within the overall structure. The authors claim to have confidently fit 39 proteins into their cryoEM maps, though there are no statistics provided. While the methods were adequately explained. For each of those proteins that were fit by sequence-based methods, it would be helpful to show examples of the fit that allowed an unambiguous call (in the supp data). For those proteins that were fit by a structure-based method, the authors seem to be on much shakier ground, as presumably the density isn't high enough resolution to call side chains even with an ambiguous alphabet. The authors need to make it more clear to the reader how they have unambiguously fit these models and what makes them sure. At the moment, everything is left to faith.

Other comments:

Overall, the manuscript is well written but very dense. It also relies on a combination of jargon from multiple fields. Partially this is unavoidable, but the authors should consider reducing field-specific jargon whenever possible to make the parasite biology more accessible to those interested in the structural biology and the structural biology a bit more accessible to the parasitologists. The former is probably the worse problem.

Introduction (e.g. Lines 44-50) – There should be more citations here. The knowledge of the composition of the conoid, APR, etc. certainly isn't only two studies.

148-160 – The authors should consider the addition of a figure model that clearly shows which synthetic pairs in the bridging complex are lethal/defective. Fig 3a could be modified to show this, and it could be added to extended data Fig 3. This would make it easier for the reader to follow the authors' logic in their mutagenesis choices.

174-178 – Because the authors do not bring up that the Apical SPMT has a similar MIP composition to the ICMTs until the next section of the paper, this section becomes very confusing. Figures 4d and 4i are set next to one another, clearly for comparison purposes. So, when someone reads the statement while examining figures 4d and 4i:

"The cryo-EM structures revealed that ICMT-1 and ICMT-2 have a nearly identical MIP arrangement (Fig. 4d), which also closely resembles that observed in our previously reported SPMT structure. The only differences are that ICMTs have TrxL1 instead of its homolog TrxL2 between protofilaments 12 and 13 (Fig. 4d, red arrows) and two additional MIPs, TLAP3 and TLAP4." The statement (while true) appears to be false. This can be solved by clarifying the statement and/or directing readers to extended figure 5c-h.

Reviewer #3 (Remarks to the Author):

In this very strong work, the authors use cryoEM single particle analysis to determine the high-resolution structure and build atomic models for the conoid and associated cytoskeletal structures isolated from the protozoan parasite, *Toxoplasma gondii*. Resolution of the structures, including tubulin-based conoid fibers, the P2 pre-conoidal ring, intraconoid microtubules, and subpellicular microtubules ranged from ~13 down to 3.4 angstroms, enabling identification of proteins from structure data. The authors identify 39 conoid-associated proteins, including several previously unknown or unstudied proteins. Localization studies using Ultrastructure Expansion Microscopy (U-ExM) validated position of newly identified proteins within the conoid of intact cells, and structure-guided mutational analysis demonstrates requirement for some of these proteins in parasite fitness and host-cell invasion. The P2 pre-conoidal ring, in particular, is quite complex and interesting.

The work is rigorous, presented well, provides numerous novel discoveries and mechanistic insights into assembly and function of the conoid – a parasite-specific structure that is critical for parasite motility and host cell invasion. Importantly, the work also provides a basis for building and testing models for conoid function. Fundamental and novel aspects of tubulin-

based structures, relevance for multiple pathogens, and detailed structural insights are expected to make the work of high interest for a broad readership. I have some few comments and questions that I think will help the authors strengthen an already very strong piece of work.

Comments/Questions:

1. Line 25 (and elsewhere): Regarding describing the CFs as open "C-shaped": the shape of the actual pfs looks more like a 'comma', or apostrophe, or perhaps a 'bent' C. the term "open C-shape" does not accurately describe the shape. While that terminology might be used in the literature, it is recommended to change it to better reflect the shape that is now clear. If one considers the MIPs associated with the PFs, perhaps the shape then looks like a backward Epsilon.
2. Line 81: re twist angle, clarify by interacting tubulin "subunits", do you mean a/b dimers or a-b interaction?
3. I find the 'seam' position to be odd. Typically, a MT seam occurs where a cylinder of pfs comes together to form a full tube. Here it appears to be between two 'sheets' of pfs. Can the authors comment on this? Likewise, the inter-protofilament contacts between the A2-A3 and A3-A4 protofilaments are quite unusual, relative to most inter-protofilament contacts in other tubulin-based filaments, as there appears to be no direct contact between the tubulins, with contacts provided by MIPs/MAPs that interdigitate between the adjacent pfs. Indeed, it seems that, rather than a contiguous sheet or tubule structure, the CF is comprised of three separate pf-containing structures. Can the authors provide some more commentary on this unusual arrangement?
4. Figures: Please adjust the arrows indicating rotation so that it is clear whether the arrow is coming out toward the viewer, or going back, away from the viewer.
5. Lines 106-107. It's unclear how prior data lead to speculation of myosin as occupying density in earlier cryoET structures but not present in the current cryoEM structure. Please provide more clear basis for this suggestion, or remove the speculation.
6. Regarding the "bridging complex". It would be helpful to provide a figure (can be suppl) indicating the relative position of adjacent CFs. e.g. something like in Fig 3a, but include something to indicate position of adjacent CFs.
7. Structure-guided analysis of CF proteins is a very nice complement to the in-depth structural data. Notably, however, the only biological read-out examined appears to be impact on growth/viability/fitness/invasion of the organism? The authors have in hand tools to readily assess the impact on conoid structure (perhaps even dynamics). The paper would be greatly strengthened if such analyses could be provided, as viability/fitness is a somewhat crude assessment of function. The conoid itself is required for host cell invasion, so it isn't surprising that some of the new conoid proteins influence invasion. What is transformative for advancing understanding is mechanistic insight into how the conoid operates.
8. Lines 148-161 (paragraph). The concept of "functional redundancy" is a bit over-emphasized and, in my view, presents an over-simplified view of protein functions. Perhaps, a better term would be "functional overlap". Indeed, I would say part of the great value of the structures determined in this work includes the point that structure argues against "functional redundancy". For example, given the distinct structure, organization, placement and interactions for each of the three proteins in the synthetic lethal/defective mutants (CF1, CIP3, and SAS6L), these proteins almost certainly are not "redundant". More likely, each protein confers specific, non-redundant functions, though there may be 'overlap' in that they each also contribute to overall stability of the CF and by extension to cell viability/functionality. I recommend rephrasing this narrative.
9. line 167. The phrasing of observing ICMT pairs "attached or detached" from the conoid is a bit confusing - it may be interpreted as indicating that within a full conoid, the ICMT might be attached to CF or not attached. But, I think the authors mean that the ICMT observed have sometimes been removed/dislodged from the entire conoid during preparation. Is that correct? Please clarify in the text.
10. Line 176-177: the text indicates SPMT have TrxL2 at pf12-13 junction, but Fig 4d shows TrxL1 in SPMT and ICMT at this position.
11. Fig 4f/g and text lines 178-182. The U-ExM images do not appear to agree with cryoEM data and text. The text and cryoEM structure are consistent with each other, with TLAP3 arcing across several pfs and differing from TLAP4, which runs along the ICMT seam. But the U-ExM images in fig 4f and g indicate TLAP3 and TLAP4 show similar distribution, as a line parallel with the conoid long axis, rather than TLAP3 running around the conoid, as indicated by cryoEM and stated in the text. Please clarify.
12. Line 206 and Fig 4. Fig 4 is missing panel 'k'.
13. Ref 38 (bioRxiv paper) appears to be superseded by a 2023 Nature Communications paper - please clarify.
14. Lines 235-236. While the cryoEM-SPA structure aligns with P2 ring from the cryoET structure, the authors should also show what alignment looks like for alignment with the P1 and P23 rings, in order to demonstrate best alignment is with P2. Can the alignments be done in a way that provides a numerical/statistical score of the quality of alignment?
15. Are the origins of the ICMTs observed? While the sample is not in situ within cells, there are some data, e.g. extended figure 5b, that indicate origins of ICMTs could be visible.

Version 3:

Decision Letter:

Our ref: NSMB-A51103C

9th Oct 2025

Dear Dr. Zhang,

Thank you for submitting your revised manuscript "Atomic models of the conoid, the cell invasion machinery of the human parasite *Toxoplasma gondii*" (NSMB-A51103C). It has now been seen by the original referees and their comments are below. The reviewers find that the paper has improved in revision, and therefore we'll be happy in principle to publish it in

Nature Structural & Molecular Biology, pending minor revisions to satisfy the referees' final requests and to comply with our editorial and formatting guidelines.

We are now performing detailed checks on your paper and will send you a checklist detailing our editorial and formatting requirements in about 2-3 weeks. Please do not upload the final materials and make any revisions until you receive this additional information from us.

Sincerely,
Kat

Katarzyna Ciazynska, PhD
(she/her)
Senior Editor
Nature Structural & Molecular Biology
<https://orcid.org/0000-0002-9899-2428>

Reviewer #1 (Remarks to the Author):

I am happy with the MS revisions that the authors have made to my comments as well as to the remarks from the two other reviewers.

Reviewer #2 (Remarks to the Author):

The authors have addressed all of my concerns about the manuscript and have improved upon an already very strong piece of science. I have no other concerns or comments.

Reviewer #3 (Remarks to the Author):

The authors have provided a thorough and orifices response to reviewer comments.

Version 4:

Decision Letter:

11th Nov 2025

Dear Dr. Zhang,

We are now happy to accept your revised paper "Atomic models of the Toxoplasma cell invasion machinery" for publication as an Article in Nature Structural & Molecular Biology.

Your paper will be published online soon after we receive proof corrections and will appear in print in the next available issue. You can find out your date of online publication by contacting the production team shortly after sending your proof corrections.

Authors may need to take specific actions to achieve compliance with funder and institutional open access mandates. If your research is supported by a funder that requires immediate open access (e.g. according to [Plan S principles](https://www.springernature.com/gp/open-science/plan-s-compliance) or the [NIH public access policy](https://www.springernature.com/gp/open-science/us-federal-agency-compliance)) then you should select the gold OA route, and we will direct you to the compliant route where possible. Because authors warrant under our subscription licensing terms that they haven't committed to licensing any version of their article under a licence inconsistent with the terms of our agreement – including the applicable embargo period – publication under the subscription model isn't suitable for authors whose funders require no embargo.

Sincerely,

Katarzyna Ciazynska, PhD
(she/her)
Senior Editor
Nature Structural & Molecular Biology
<https://orcid.org/0000-0002-9899-2428>

We thank the reviewers for their comments, which have helped improve the paper. During the revision we have added new data and figures in response to the reviewer's comments and questions. These additions include:

1. A new panel Fig. 1f showing the inter-protofilament angle measurements of the conoid fiber.
2. Updated Fig. 3a to include the atomic model of the adjacent CF.
3. An inset in Fig. 3b to indicate the synthetic lethal/defective pairs.
4. Moved the original Extended Data Fig. 5 to be the new Fig. 5 focusing on the apical SPMTs.
5. A new Supplementary Table 2 to summarize the conservation of all identified conoid-associated proteins across apicomplexan parasites.
6. A new Extended Data Fig. 2 to demonstrate our protein identification workflow, with examples of both sequence- and structure-based approaches.
7. A new Extended Data Fig. 5, in which we use transmission electron microscopy (TEM) to show that the synthetic lethal/defective pairs of CF proteins disrupt the conoid structure in both intracellular and extracellular parasites.
8. New panels a-c in Extended Data Fig. 7 showing that the dimensions of the P2-SPA structure are incompatible with those of the P1 and P3 rings from cryo-ET studies.
9. We further improved the local resolution of ICMT-2 and could assign part of the MAP densities as ICMAP4 (TGME49_225340) based on side-chain information.
10. We have deposited the cryo-EM structures to the Electron Microscopy Data Bank (EMDB), and corresponding atomic models to the Protein Data Bank (PDB). We have also deposited the M/S data to the PRIDE repository. Accession numbers have been added to the Data Availability statement. Our maps/models are also immediately accessible at Figshare.

Below we provide a point-by-point response to their comments. We have also highlighted new text and citations in the revised manuscript in blue.

Reviewers' Comments:

Reviewer #1 (Remarks to the Author):

This outstanding manuscript uses cryo-EM and single-particle analysis to define the fine details of several protein complexes from the *Toxoplasma* conoid, an unusual organelle that is required for parasite motility and invasion. After determining protein complexes, the authors screen interacting proteins as an elegant strategy to identify important but functionally redundant protein pairs. The project is significant, and the data are of high quality. The manuscript is clear and complete. This paper has an extraordinary amount of important information, and I only have a few corrections or clarifications to suggest.

We thank the reviewer for the enthusiastic and encouraging comments.

Small correction: In the figure 1 D legend, please define A as it related to protofilaments (it could be interpreted as alpha).

We apologize for the confusion. 'A' refers to the A-tubule, as named in the context of the doublet microtubule. We agree that this terminology can be confusing and have removed the prefix 'A' from the numbering in Figs. 1-3 and the text to align with previous literature.

Line 41-2: "This stability is mainly attributed to the binding of microtubule inner proteins (MIPs)" –add ref 33 here because it demonstrates loss of stability when SPM1 is deleted (Tran et al).
Done.

Addition: "Given the structural conservation of these conoid components, our structures also facilitate functional studies of conoid proteins in other apicomplexans, including *Cryptosporidium* and *Plasmodium* spp."

It would be valuable to add one more column to table 1. For each component what is the degree of conservation across the phylum? That is, are they in all apicomplexans, in the coccidia, in other alveolates, etc. Given differences in the appearance of the conoid in the Haemosporida, I would hypothesize that some of the conoid fiber components would be lost in *Plasmodium* etc.

We thank the reviewer for the suggestion. We have added a new Supplementary Table 2 to summarize the conservation of all identified conoid-associated proteins across apicomplexan parasites.

Questions:

1. There are two unconventional alpha tubulins in the *Toxoplasma* genome that have an insert in the N loop (TGME49_231770) or an unusually long carboxyterminal tail and missing GAP domain residues (TGME49_231400). Can either of these be placed in the structure, particularly in areas of greater structural torsion (ie between protofilaments A3 and A4)?

We thank the reviewer for this excellent question. We have carefully checked the side chain densities of our cryo-EM structure and concluded that all the α -tubulin in conoid fibers are the α 1 isoform (TGME49_316400) rather than α 2 (TGME49_231770) or α 3 (TGME49_231400). In addition, our M/S analysis only identified α 1-tubulin. We have added the result to the main text.

2. TLAP3 is an arc-MIP which spans laterally across 11 out of the 13 protofilaments and inserts residues into multiple taxol-binding pockets located at different protofilaments...Can you expand upon how TLAP3 is similar or different from *Chlamydomonas* flagellar arc-MIPs?

The microtubule-binding mode of TLAP3 indeed closely resembles that of the arc-MIP first described in *Chlamydomonas* central pair microtubules. We have added this statement to the main text.

3. It is surprising to me that there are so many novel proteins in these complexes, given the repurposing of SFA and SAS6L in the conoid. I am surprised to not see DIP13, a homolog of a *Chlamydomonas* flagellar apparatus component that is localized to the conoid appear in this description of key proteins. Could you comment on why this may be?

DIP13 (TGME49_295450) ranked #805 in our M/S result but was not found in our cryo-EM structure. Notably, DIP13 has also not been observed as a structural component of the doublet microtubules or central pair microtubules in *Chlamydomonas* flagella. Its absence could be due

to two possibilities: (1) DIP13 is a structural component of the CF but does not follow the 8-nm tubulin periodicity and is therefore averaged out in the cryo-EM structure; or (2) DIP13 is a mobile element that binds to the CF transiently.

Reviewer #2 (Remarks to the Author):

One of the hallmarks of apicomplexan parasites is the conserved apical complex that is essential for motility, invasion, and effector secretion. Recent advances in cryoEM and sample preparation for light microscopy have enabled a rapid increase in our understanding of this fascinating structure. In the present paper, the authors adapted a previously published protocol for the purification of the detergent-stable components of the Toxoplasma apical complex cytoskeleton, including the conoid and its associated microtubules. The authors averaged the cytoskeleton fibers to produce reasonably high resolution maps that allowed them to reliably identify a number of proteins known to be associated with the complex. While the apical complex is reasonably robust to loss of single proteins, the authors' structural models enabled them to predict double-mutations that would show synthetic lethality. Taken together, this really is a tour-de-force.

We thank the reviewer for the enthusiastic and encouraging comments.

My comments focus on data availability, clarifying methods and model quality, and some changes to make the manuscript more accessible to a broad audience.

Major comments:

Data availability: It is frustrating that there are no deposition identifiers for any of the data (MS, cryoEM maps, or models), suggesting that the authors have not deposited any of these datasets. Given that all repositories allow embargoes there really isn't any reason for this. It's doubly confounding given that the PDB reports they provided explicitly say "not for manuscript review". This means the quality of the models and maps cannot be reliably evaluated by a reviewer. While I trust the authors as scientists, this is a bad precedent, and I thought was against NSMB policy.

We apologize for not depositing the maps/models prior to the first submission. The only reason is that we anticipated that our maps/models quality might continue to improve during the extended revision process. We have now deposited all the maps/models to EMDB/PDB. We have also deposited the M/S data to the PRIDE repository. Accession numbers have been added to the Data Availability statement. Our maps/models are also immediately accessible at Figshare.

The major importance of this work is the placement of specific proteins within the overall structure. The authors claim to have confidently fit 39 proteins into their cryoEM maps, though there are no statistics provided. While the methods were adequately explained · For each of those proteins that were fit by sequence-based methods, it would be helpful to show examples of the fit that allowed an unambiguous call (in the supp data). For those proteins that were fit

by a structure-based method, the authors seem to be on much shakier ground, as presumably the density isn't high enough resolution to call side chains even with an ambiguous alphabet. The authors need to make it more clear to the reader how they have unambiguously fit these models and what makes them sure. At the moment, everything is left to faith.

We thank the reviewer for raising this important concern. We have added a new Extended Data Fig. 2 to illustrate our protein identification workflow, with examples of both sequence- and structure-based approaches. For both methods, we are fully confident in the assignments. In the structure-based approach, we only assign proteins whose AlphaFold-predicted domain structure essentially matched the densities in the spatial arrangement of secondary structural elements (α -helices and β -sheets). For globular proteins, this method has been shown to provide results as accurate as the sequence-based approach, as demonstrated by direct comparison (e.g. Chen, Z. *et al. Cell* 2023 and Leung, Z. *et al. Cell* 2023).

Other comments:

Overall, the manuscript is well written but very dense. It also relies on a combination of jargon from multiple fields. Partially this is unavoidable, but the authors should consider reducing field-specific jargon whenever possible to make the parasite biology more accessible to those interested in the structural biology and the structural biology a bit more accessible to the parasitologists. The former is probably the worse problem.

We thank the reviewer for pointing out this issue. We have gone through the text and removed or explained the jargon as much as possible within the word count limit of the journal.

Introduction (e.g. Lines 44-50) – There should be more citations here. The knowledge of the composition of the conoid, APR, etc. certainly isn't only two studies.

We have added five additional references.

148-160 – The authors should consider the addition of a figure model that clearly shows which synthetic pairs in the bridging complex are lethal/defective. Fig 3a could be modified to show this, and it could be added to extended data Fig 3. This would make it easier for the reader to follow the authors' logic in their mutagenesis choices.

We have updated Fig. 3a to include the atomic model of the adjacent CF and added an inset in Fig. 3b to indicate the synthetic lethal/defective pairs.

174-178 – Because the authors do not bring up that the Apical SPMT has a similar MIP composition to the ICMTs until the next section of the paper, this section becomes very confusing. Figures 4d and 4i are set next to one another, clearly for comparison purposes. So, when someone reads the statement while examining figures 4d and 4i:

“The cryo-EM structures revealed that ICMT-1 and ICMT-2 have a nearly identical MIP arrangement (Fig. 4d), which also closely resembles that observed in our previously reported SPMT structure. The only differences are that ICMTs have TrxL1 instead of its homolog TrxL2 between protofilaments 12 and 13 (Fig. 4d, red arrows) and two additional MIPs, TLAP3 and

TLAP4." The statement (while true) appears to be false. This can be solved by clarifying the statement and/or directing readers to extended figure 5c-h.

We agree with the reviewer that the original wording was confusing and have rephrased this paragraph: " The cryo-EM structures revealed that ICMT-1 and ICMT-2 share an almost identical MIP arrangement (Fig. 4d), which is also the same as that of the apical subpellicular microtubules (SPMTs) (see the next section). The MIPs in these three microtubule types closely resemble those in our previously reported central SPMT structure (Fig. 5f,g), with two subtle differences: (i) TrxL1 replaces its homolog TrxL2 between protofilaments 12 and 13 (Figs. 4d,5f, red arrows), and (ii) two additional MIPs, TLAP3 and TLAP4, are present (Figs. 4e,5g, red arrows)."

Reviewer #3 (Remarks to the Author):

In this very strong work, the authors use cryoEM single particle analysis to determine the high-resolution structure and build atomic models for the conoid and associated cytoskeletal structures isolated from the protozoan parasite, *Toxoplasma gondii*. Resolution of the structures, including tubulin-based conoid fibers, the P2 pre-conoidal ring, intraconoid microtubules, and subpellicular microtubules ranged from ~13 down to 3.4 angstroms, enabling identification of proteins from structure data. The authors identify 39 conoid-associated proteins, including several previously unknown or unstudied proteins. Localization studies using Ultrastructure Expansion Microscopy (U-ExM) validated position of newly identified proteins within the conoid of intact cells, and structure-guided mutational analysis demonstrates requirement for some of these proteins in parasite fitness and host-cell invasion. The P2 preconoidal ring, in particular, is quite complex and interesting.

The work is rigorous, presented well, provides numerous novel discoveries and mechanistic insights into assembly and function of the conoid – a parasite-specific structure that is critical for parasite motility and host cell invasion. Importantly, the work also provides a basis for building and testing models for conoid function. Fundamental and novel aspects of tubulin-based structures, relevance for multiple pathogens, and detailed structural insights are expected to make the work of high interest for a broad readership. I have some few comments and questions that I think will help the authors strengthen an already very strong piece of work.

We thank the reviewer for the enthusiastic and encouraging comments.

Comments/Questions:

1. Line 25 (and elsewhere): Regarding describing the CFs as open "C-shaped": the shape of the actual pfs looks more like a 'comma', or apostrophe, or perhaps a 'bent' C. the term "open C-shape" does not accurately describe the shape. While that terminology might be used in the literature, it is recommended to change it to better reflect the shape that is now clear. If one considers the MIPs associated with the PFs, perhaps the shape then looks like a backward Epsilon.

We agree with the reviewer and have changed the "C-shaped" to "bent C-shaped".

2. Line 81: re twist angle, clarify by interacting tubulin "subunits", do you mean a/b dimers or a-b interaction?

It refers to the twist angle between one α,β -tubulin heterodimer and the next heterodimer within the same protofilament. We have added this clarification to the text.

3. I find the 'seam' position to be odd. Typically, a MT seam occurs where a cylinder of pfs comes together to form a full tube. Here it appears to be between two 'sheets' of pfs. Can the authors comment on this? Likewise, the inter-protofilament contacts between the A2-A3 and A3-A4 protofilaments are quite unusual, relative to most inter-protofilament contacts in other tubulin-based filaments, as there appears to be no direct contact between the tubulins, with contacts provided by MIPs/MAPs that interdigitate between the adjacent pfs. Indeed, it seems that, rather than a contiguous sheet or tubule structure, the CF is comprised of three separate pf-containing structures. Can the authors provide some more commentary on this unusual arrangement?

We have added a new panel Fig. 1f showing the inter-protofilament angle measurements of the conoid fiber. The three MIPs (CF2, CF4 and CF3) appear to delineate the boundaries of three distinct PF groups (1-2, 3-4, 5-9). The seam angle between PFs 4 and 5 (31.8°) is relatively close to the canonical inter-protofilament angle of 13-PF microtubule (27.7°). The most unusual angles occur between PFs 2-3 (50.8°) and 3-4 (73.4°), where canonical tubulin lateral interfaces are preserved but further stabilized by neighboring microtubule associated proteins. We have incorporated more description into the main text.

4. Figures: Please adjust the arrows indicating rotation so that it is clear whether the arrow is coming out toward the viewer, or going back, away from the viewer.

We have added a solid dot (\bullet) next to the arrows to indicate that the arrow is coming out toward the viewer.

5. Lines 106-107. It's unclear how prior data lead to speculation of myosin as occupying density in earlier cryoET structures but not present in the current cryoEM structure. Please provide more clear basis for this suggestion, or remove the speculation.

We agree that the current structural data (cryo-EM and cryo-ET) do not provide sufficient evidence to assign the extra density to myosin H; therefore, we have removed this speculation.

6. Regarding the "bridging complex". It would be helpful to provide a figure (can be suppl) indicating the relative position of adjacent CFs. e.g. something like in Fig 3a, but include something to indicate position of adjacent CFs.

We have updated Fig.3a to include the atomic model of the adjacent CF.

7. Structure-guided analysis of CF proteins is a very nice complement to the in-depth structural data. Notably, however, the only biological read-out examined appears to be impact on growth/viability/fitness/invasion of the organism? The authors have in hand tools to readily assess the impact on conoid structure (perhaps even dynamics). The paper would be greatly strengthened if such analyses could be provided, as viability/fitness is a somewhat crude

assessment of function. The conoid itself is required for host cell invasion, so it isn't surprising that some of the new conoid proteins influence invasion. What is transformative for advancing understanding is mechanistic insight into how the conoid operates.

We thank the reviewer for this great suggestion. We have added a new Extended Data Fig. 5, in which we use transmission electron microscopy to show that the synthetic lethal/defective pairs of CF proteins substantially disrupt the conoid structure in both intracellular and extracellular parasites.

8. Lines 148-161 (paragraph). The concept of "functional redundancy" is a bit over-emphasized and, in my view, presents an over-simplified view of protein functions. Perhaps, a better term would be "functional overlap". Indeed, I would say part of the great value of the structures determined in this work includes the point that structure argues against "functional redundancy". For example, given the distinct structure, organization, placement and interactions for each of the three proteins in the synthetic lethal/defective mutants (CF1, CIP3, and SAS6L), these proteins almost certainly are not "redundant". More likely, each protein confers specific, non-redundant functions, though there may be 'overlap' in that they each also contribute to overall stability of the CF and by extension to cell viability/functionality. I recommend rephrasing this narrative.

We completely agree with the reviewer and have revised the paragraph regarding "functional overlap".

9. line 167. The phrasing of observing ICMT pairs "attached or detached" from the conoid is a bit confusing - it may be interpreted as indicating that within a full conoid, the ICMT might be attached to CF or not attached. But, I think the authors mean that the ICMT observed have sometimes been removed/dislodged from the entire conoid during preparation. Is that correct? Please clarify in the text.

Yes, the ICMT observed have sometimes been removed/dislodged from the entire conoid during preparation. We have modified the text accordingly.

10. Line 176-177: the text indicates SPMT have TrxL2 at pf12-13 junction, but Fig 4d shows TrxL1 in SPMT and ICMT at this position.

We apologize for the confusion. The apical SPMT and both ICMTs have TrxL1 between PF12 and 13 (this study), whereas the central SPMT has TrxL2 at the same location (Wang et al. Nat Commun 2021). We are confident in both assignments. To clarify the resemblances and differences, we have revised this paragraph, which was also mentioned by reviewer #2.

11. Fig 4f/g and text lines 178-182. The U-ExM images do not appear to agree with cryoEM data and text. The text and cryoEM structure are consistent with each other, with TLAP3 arcing across several pfs and differing from TLAP4, which runs along the ICMT seam.. But the U-ExM images in fig 4f and g indicate TLAP3 and TLAP4 show similar distribution, as a line parallel with the conoid long axis, rather than TLAP3 running around the conoid, as indicated by cryoEM and stated in the text. Please clarify.

We apologize for the confusion. The cryo-EM structure shown in Fig. 4e is one microtubule, with TLAP3 binding across several protofilaments. In contrast, the U-ExM images in Fig. 4f-g show one conoid above multiple subpellicular microtubules, which is at a much larger scale. To clarify it, we have added a scale bar to Fig. 4e.

12. Line 206 and Fig 4. Fig 4 is missing panel 'k'.

Panel 4k was in the top right corner. We apologize for the original layout of Fig. 4. To avoid confusion, we have merged the panels of apical SPMTs with the original Extended Data Fig. 5 and created a new Fig. 5 focusing on the apical SPMTs.

13. Ref 38 (bioRxiv paper) appears to be superseded by a 2023 Nature Communications paper - please clarify.

We have updated the reference.

14. Lines 235-236. While the cryoEM-SPA structure aligns with P2 ring from the cryoET structure, the authors should also show what alignment looks like for alignment with the P1 and P23 rings, in order to demonstrate best alignment is with P2. Can the alignments be done in a way that provides a numerical/statistical score of the quality of alignment?

We were unable to achieve a reasonable docking (with numerical correlation score) of the SPA structure into the cryo-ET densities of the P1 and P3 rings using the *'fit inMap'* command in ChimeraX software, indicating a poor match. In contrast, docking the SPA structure into the cryo-ET density of the P2 ring yielded a stable solution, with a decent correlation score of 0.7838. To illustrate this, we have added new panels a-c in Extended Data Fig. 7, showing that the shape and dimensions of the P2-SPA structure are incompatible with those of the P1 and P3 rings.

15. Are the origins of the ICMTs observed? While the sample is not in situ within cells, there are some data, e.g. extended figure 5b, that indicate origins of ICMTs could be visible.

We attempted to reconstruct the end structures of SPMTs (minus ends) and ICMTs, but without success. Only a limited number of ends were captured in our micrographs (e.g., Fig. 4b), and they displayed substantial structural heterogeneity, making them unsuitable for averaging.